# The SARS-CoV-2 accessory protein Orf3a is not an ion channel, but does interact with trafficking proteins

**Alexandria N Miller[1]\*[†], Patrick R Houlihan[1†], Ella Matamala[2], Deny Cabezas-Bratesco[2], Gi Young Lee[3‡], Ben Cristofori-Armstrong[1§], Tanya L Dilan[1], Silvia Sanchez-Martinez[1#], Doreen Matthies[1¶], Rui Yan[1], Zhiheng Yu[1], Dejian Ren[3], Sebastian E Brauchi[1,2], David E Clapham[1]\***

[1]Janelia Research Campus, Ashburn, United States; [2]Physiology Institute and Millennium Nucleus of Ion Channel-Associated Diseases, Universidad Austral de Chile, Valdivia, Chile; [3]Department of Biology, University of Pennsylvania, Philadelphia, United States

**\*For correspondence:**
millera@janelia.hhmi.org (ANM);
claphamd@hhmi.org (DEC)

[†]These authors contributed equally to this work

**Present address:** [‡]Department of Microbiology and Immunology, Cornell University, Ithaca, United States; [§]Center for Advanced Imaging, University of Queensland, St. Lucia, Australia; [#]Molecular Biology Department, University of Wyoming, Laramie, United States; [¶]Unit on Structural Biology, Division of Basic and Translational Biophysics, Eunice Kennedy Shriver National Institute of Child Health and Human Development, National Institutes of Health, Bethesda, United States, United States

**Competing interest:** The authors declare that no competing interests exist.

**Abstract** The severe acute respiratory syndrome associated coronavirus 2 (SARS-CoV-2) and SARS-CoV-1 accessory protein Orf3a colocalizes with markers of the plasma membrane, endocytic pathway, and Golgi apparatus. Some reports have led to annotation of both Orf3a proteins as viroporins. Here, we show that neither SARS-CoV-2 nor SARS-CoV-1 Orf3a form functional ion conducting pores and that the conductances measured are common contaminants in overexpression and with high levels of protein in reconstitution studies. Cryo-EM structures of both SARS-CoV-2 and SARS-CoV-1 Orf3a display a narrow constriction and the presence of a positively charged aqueous vestibule, which would not favor cation permeation. We observe enrichment of the late endosomal marker Rab7 upon SARS-CoV-2 Orf3a overexpression, and co-immunoprecipitation with VPS39. Interestingly, SARS-CoV-1 Orf3a does not cause the same cellular phenotype as SARS-CoV-2 Orf3a and does not interact with VPS39. To explain this difference, we find that a divergent, unstructured loop of SARS-CoV-2 Orf3a facilitates its binding with VPS39, a HOPS complex tethering protein involved in late endosome and autophagosome fusion with lysosomes. We suggest that the added loop enhances SARS-CoV-2 Orf3a's ability to co-opt host cellular trafficking mechanisms for viral exit or host immune evasion.

## Editor's evaluation

The function of specific proteins made by SARS-CoV-1 and SARS-CoV-2 is under debate, with diverging claims previously published regarding the ability of Orf3a proteins from either virus to form ion channels. The authors undertook a thorough characterization of Orf3a from CoV-1 and CoV-2 by combining data from a range of different structural and functional experiments, arguably providing the most compelling evidence to date that Orf3a from viruses is not an ion channel. Instead, the orthologue-specific interaction with a component of a larger protein complex suggests the role of one of the two membrane proteins in the endo-lysosomal pathway. The work is significant from a fundamental science perspective, for its implications for COVID antiviral development strategies, and also for establishing guidelines for future identification of true viral ion channels.

## Introduction

The β-coronavirus, Severe Acute Respiratory Syndrome CoronaVirus 2 (SARS-CoV-2) has among the largest genomes of any RNA virus (*Bar-On et al., 2020*; *Hartenian et al., 2020*), encoding for 29 proteins. The majority of these proteins are 'Non-Structural Proteins' (NSPs) that mediate viral RNA replication, while the 'structural proteins' are components of the virion (*Bar-On et al., 2020*; *Hartenian et al., 2020*). The third group of functionally enigmatic 'accessory' proteins likely bolster SARS-CoV-2 replication or facilitate evasion from the host's innate immune system. One of these proteins is Open reading frame 3a (Orf3a), a membrane protein that is annotated as a putative viroporin based on previous work and its similarity to SARS-CoV-1, also claimed to be a viroporin (*Kern et al., 2021*; *Toft-Bertelsen et al., 2021*; *Lu et al., 2006*; *Chan et al., 2009*). In this paper, we determine the structures of SARS-CoV-2 and SARS-CoV-1 Orf3a, measure their functions, and examine the differences between the CoV-2 and CoV-1 proteins.

Viroporins are viral membrane proteins that commonly form weakly- or non-selective oligomeric holes in surface plasma membranes or intracellular organellar membranes (*Nieva et al., 2012*; *Scott and Griffin, 2015*). SARS-CoV-1 Orf3a was postulated to be a viroporin based on its membrane localization, oligomerization, and C-terminal sequence similarities to a calcium ATPase from *Plasmodium falciparum* and an outer-membrane porin from *Shewanella oneidensis* (*Lu et al., 2006*; *Yu et al., 2004*). Recording $K^+$-selective currents from SARS-CoV-1 Orf3a-expressing *Xenopus laevis* oocytes were interpreted to mean that SARS-CoV-1 Orf3a was a viroporin (*Lu et al., 2006*). This current was inhibited by 5–50 μM emodin, a natural compound suggested as a potential SARS-CoV-1 antiviral (*Lu et al., 2006*; *Schwarz et al., 2011*), and by 10 mM $BaCl_2$, a nonspecific $K^+$ channel blocker. Whole-cell patch-clamp of SARS-CoV-1 expressing HEK293 cells exhibited non-selective ion currents blocked by $BaCl_2$ that were also attributed to SARS-CoV-1 Orf3a (*Chan et al., 2009*). Although these data were used to support the SARS-CoV-1 Orf3a viroporin hypothesis, further work to identify its pore-lining residues is lacking (*Lu et al., 2006*; *Chan et al., 2009*). One study pursued this question by testing SARS-CoV-1 Orf3a pore mutants in artificial bilayers (*Castaño-Rodriguez et al., 2018*). However, this method is prone to background channel contamination and the validity of this observation has been recently questioned (*Accardi et al., 2004*; *McClenaghan et al., 2020*).

Comparison of viral infections between wild-type and Orf3a-deficient SARS-CoV-1 strains demonstrates that Orf3a promotes host cell death and causes intracellular vesicle formation and Golgi fragmentation (*Freundt et al., 2010*). In addition, during SARS-CoV-1 infection, Orf3a stimulates the production of mature IL-1β, a marker of NLRP3 inflammasome activation (*Siu et al., 2019*). Although viroporins from other viruses have been shown to trigger apoptosis or innate immune cell activation, many other viral hijacking mechanisms can produce these phenotypes (*Nieva et al., 2012*; *Scott and Griffin, 2015*). The lack of compelling evidence connecting the contributions of Orf3a to SARS-CoV-1 pathogenesis with its purported viroporin activity raises the question of whether its channel function is physiologically required and if SARS-CoV-1 Orf3a is a *bona fide* viroporin.

Given its similarity to the SARS-CoV-1 homolog, we asked if SARS-CoV-2 Orf3a is a viroporin. While pursuing this, several groups reported conflicting findings that support or oppose the SARS-CoV-2 Orf3a viroporin hypothesis (*Kern et al., 2021*; *Toft-Bertelsen et al., 2021*; *Grant and Lester, 2021*). We reasoned that the discrepancy could be due to common experimental problems that arise when characterizing putative ion channels, leading to its misidentification as a viroporin (*Accardi et al., 2004*; *Harrison et al., 2022*; *Niu et al., 2021*). Additionally, SARS-CoV-2 Orf3a viroporin activity has not been studied in mammalian cells, which represents a more native environment for functional studies (*Kern et al., 2021*; *Toft-Bertelsen et al., 2021*; *Grant and Lester, 2021*). We performed a comprehensive structural and functional investigation of SARS-CoV-2 Orf3a. We surveyed the subcellular localization of overexpressed SARS-CoV-2 Orf3a in HEK293 cells and observed co-localization with markers of the plasma membrane and the endocytic pathway, as previously reported (*Ghosh et al., 2020*; *Gordon et al., 2020a*; *Zhang et al., 2020*; *Miao et al., 2021*). We then made extensive efforts to record SARS-CoV-2 Orf3a attributable cation currents at the plasma membrane and in endolysosomes of HEK293 cells. We also attempted to measure SARS-CoV-2 Orf3a currents at the plasma membrane of A549 lung alveolar cells (a model of alveolar type II pneumocytes that are a target host cell of SARS-CoV-2 infection), in *Xenopus* oocytes, and in a reconstituted system. In all cell lines and reconstituted systems tested, we did not measure currents attributable to SARS-CoV-2 Orf3a. We explored this further by resolving three high-resolution cryo-EM structures of SARS-CoV-2 Orf3a under different conditions,

varying the lipid composition and the scaffold protein used for nanodisc assembly. All structures were captured in the same conformational state, displaying a constriction within the transmembrane region and the presence of a positively charged aqueous vestibule, which would not favor cation permeation. We were also unable to reproduce the published SARS-CoV-1 Orf3a recordings from *Xenopus* oocytes and HEK293 cells. Finally, we show that the SARS-CoV-1 Orf3a cryo-EM structure mirrors the overall architecture and structural features seen in the SARS-CoV-2 homolog. From our data, we conclude that SARS-CoV-1 and SARS-CoV-2 Orf3a are not viroporins.

What is the function of SARS-CoV-2 Orf3a and how may it contribute to SARS-CoV-2 pathogenicity? We observe Rab7 enrichment, a marker for late endosomes, upon SARS-CoV-2 Orf3a overexpression, and co-immunoprecipitation with the host protein, VPS39. In contrast, SARS-CoV-1 Orf3a does not cause the same cellular phenotype and does not interact with VPS39. We identified an unstructured, cytosolic loop unique to SARS-CoV-2 Orf3a that contributes to its interaction with VPS39. Our data is in agreement with previous work showing that SARS-CoV-2 Orf3a binds VPS39, a HOPS complex tethering protein involved in autophagosome and late endosome fusion with lysosomes (*Miao et al., 2021*; *Chen et al., 2021*; *Qu et al., 2021*; *Balderhaar and Ungermann, 2013*). Disruption of HOPS complex activity may be a mechanism to promote SARS-CoV-2 viral egress, an exit strategy recently proposed for β-coronaviruses (*Ghosh et al., 2020*; *Chen et al., 2021*).

## Results

### SARS-CoV-2 Orf3a colocalizes with cellular markers of the plasma membrane and the endocytic pathway

SARS-CoV-1 Orf3a is enriched in the Golgi apparatus and plasma membrane, but is also present in late endosomes, lysosomes, and the perinuclear region when expressed in epithelial, fibroblast and osteosarcoma immortalized cell lines (*Chan et al., 2009*; *Yu et al., 2004*; *Freundt et al., 2010*; *Tan et al., 2004*; *Ito et al., 2005*; *Yue et al., 2018*; *Padhan et al., 2007*). To determine whether SARS-CoV-2 Orf3a displays a similar subcellular distribution as SARS-CoV-1 Orf3a, we generated doxycycline-inducible HEK293 cell lines which stably expressed SARS-CoV-2 or SARS-CoV-1 Orf3a fused to a HALO tag. Cells not treated with doxycycline served as negative controls (*Figure 1—figure supplement 1B*, *Figure 1—figure supplement 2B*). To assess localization of SARS-CoV-2 and SARS-CoV-1 Orf3a, cells were transfected or immunostained with various subcellular markers of the endoplasmic reticulum (ER), Golgi apparatus (G), plasma membrane (PM), early endosomes (EE), late endosomes (LE), lysosomes (Lyso), and peroxisomes (PX). After 24 h expression, SARS-CoV-2 Orf3a-$_{HALO}$ colocalized with the PM marker, farnesylated-GFP, which was further supported by total internal reflection microscopy (*Figure 1A–C*). We also observe partial colocalization of SARS-CoV-2 Orf3a$_{HALO}$ with markers of the endocytic pathway, including EE (EEA, Rab5), LE (Rab7) and the Lyso (LAMP1) compartments (*Figure 1D–F*, *Figure 1—figure supplement 1A, C-E*). Minimal SARS-CoV-2 Orf3a is seen in all other subcellular compartments tested (*Figure 1G–I*, *Figure 1—figure supplement 1F*) consistent with other reports (*Ghosh et al., 2020*; *Gordon et al., 2020a*; *Zhang et al., 2020*; *Miao et al., 2021*). For SARS-CoV-1 Orf3a$_{HALO}$, we identify a similar trend of colocalization with markers of the endocytic pathway and at the plasma membrane (*Figure 1—figure supplement 2*).

### SARS-CoV-2 Orf3a currents are not observed across cell or late endosome/lysosome membranes

SARS-CoV-1 Orf3a was reported to form viroporins at plasma membranes (*Lu et al., 2006*; *Schwarz et al., 2011*). Given its sequence similarity to SARS-CoV-1 Orf3a, we asked whether SARS-CoV-2 Orf3a may also exhibit similar ion channel properties. We generated a SARS-CoV-2 Orf3a$_{SNAP}$ doxycycline-inducible HEK293 cell line and performed whole-cell patch-clamp electrophysiology. In all external cationic bath solutions that we tested, including $K^+$, $Na^+$, $Cs^+$, N-methyl-D-glucamine ($NMDG^+$), and $Ca^{2+}$, we were not able to observe ionic current densities distinct from those measured in control cells (*Figure 2A–C*). We also performed whole-endolysosomal patch-clamp electrophysiology in HEK293 cells expressing SARS-CoV-2 Orf3a$_{HALO}$. No distinct $K^+$, $Na^+$, $Ca^{2+}$, or $H^+$ currents were recorded as compared to those measured from untransfected endolysosomal vesicles (*Figure 2D–I*, *Figure 2—figure supplement 1*). Our interrogation of SARS-CoV-2 Orf3a overexpressed at the PM and in the LE/Lyso of HEK293 cells suggests that SARS CoV-2 Orf3a is not forming a cation permeable viroporin.

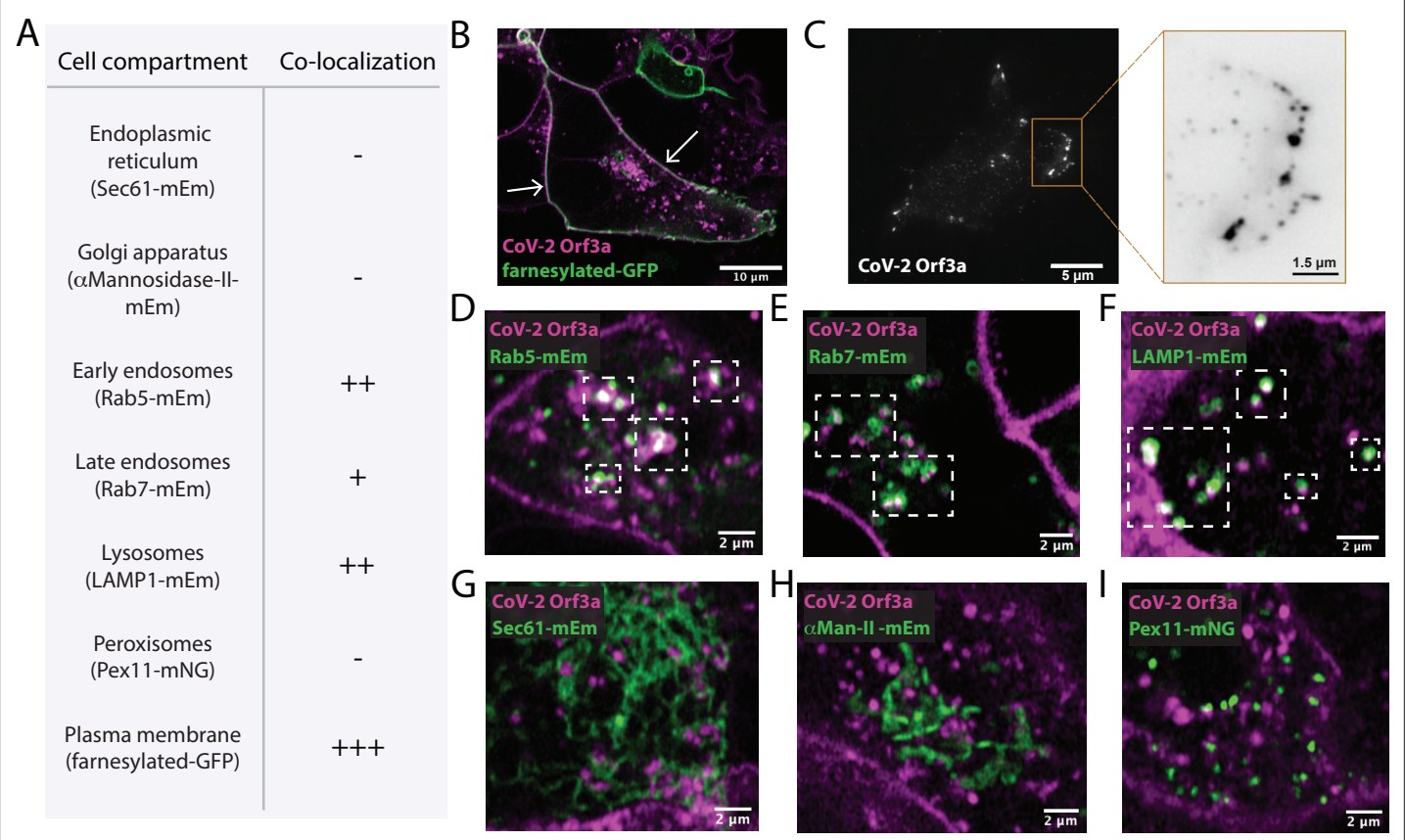

**Figure 1.** SARS-CoV-2 Orf3a colocalizes with markers for the plasma membrane and the endocytic pathway by live-cell imaging. (**A**) Summary table of SARS-CoV-2 (CoV-2) Orf3a_HALO colocalization with subcellular protein markers. All markers used to identify cellular compartments are listed in the table in A and are transiently expressed (mEm, mEmerald; mNG, mNeonGreen; GFP, green fluorescent protein). (**B**) Live-cell image of transiently expressed farnesylated-GFP (green) and CoV-2 Orf3a_HALO (magenta) using a HEK293 doxycycline-inducible CoV-2 Orf3a_HALO stable cell line. White arrows indicate co-localization. (**C**) Total Internal Reflection Fluorescence (TIRF) imaging of HEK293 cell with transient expression of CoV-2 Orf3a_HALO (white). Orange box, magnification of the surface to highlight CoV-2 Orf3a_HALO (black). (**D–I**) Live-cell image of transiently expressed (**D**) Rab5-mEm, (**E**) Rab7-mEm, (**F**) LAMP1-mEm, (**G**) Sec61-mEm, (**H**) αMannosidase-II-mEm, or (**I**) Pex11-mNG (green) with CoV-2 Orf3a_HALO (magenta) as described in (**B**). White boxes indicate regions of co-localization. All confocal images are representative of three to six independent experiments.

The online version of this article includes the following figure supplement(s) for figure 1:

**Figure supplement 1.** SARS-CoV-2 Orf3a colocalizes with markers of the endocytic pathway, but not with a Golgi marker, by immunostaining.

**Figure supplement 2.** SARS-CoV-1 Orf3a colocalizes with markers for the plasma membrane and the endocytic pathway by live-cell imaging.

## SARS-CoV-2 Orf3a currents are not observed in human alveolar cells or in *Xenopus* oocytes

Alveolar epithelial type II cells are a main target of SARS-CoV-2 viral infection. To evaluate whether the cell line used to express SARS-CoV-2 Orf3a may impact its channel activity, we performed similar experiments using human alveolar basal epithelial A549 cells. We generated a SARS-CoV-2 Orf3a_GFP doxycycline-inducible A549 cell line and performed whole-cell patch-clamp electrophysiology. SARS-CoV-2 Orf3a_GFP localizes to the plasma membrane of A549 cells, yet we were unable to measure ion currents (*Figure 2—figure supplement 2*). We next asked whether we could mimic the conditions used to characterize SARS-CoV-1 Orf3a in *Xenopus* oocytes to investigate SARS-CoV-2 Orf3a channel activity. We generated SARS-CoV-2 Orf3a_2x-STREP cRNA and injected 20 ng of cRNA or water as a control into defolliculated *Xenopus* oocytes. After 48–72 h, oocytes were recorded by two-electrode voltage clamp (TEVC) using a high potassium solution (96 mM KCl), similar to an extracellular solution that previously elicited SARS-CoV-1 Orf3a ionic currents, or standard TEVC recording solutions (ND96, 96 mM NaCl) (*Lu et al., 2006*). We were not able to record SARS-CoV-2 Orf3a_2x-STREP ionic currents above background (*Figure 2J–L*, *Figure 2—figure supplement 3*), despite confirmation of

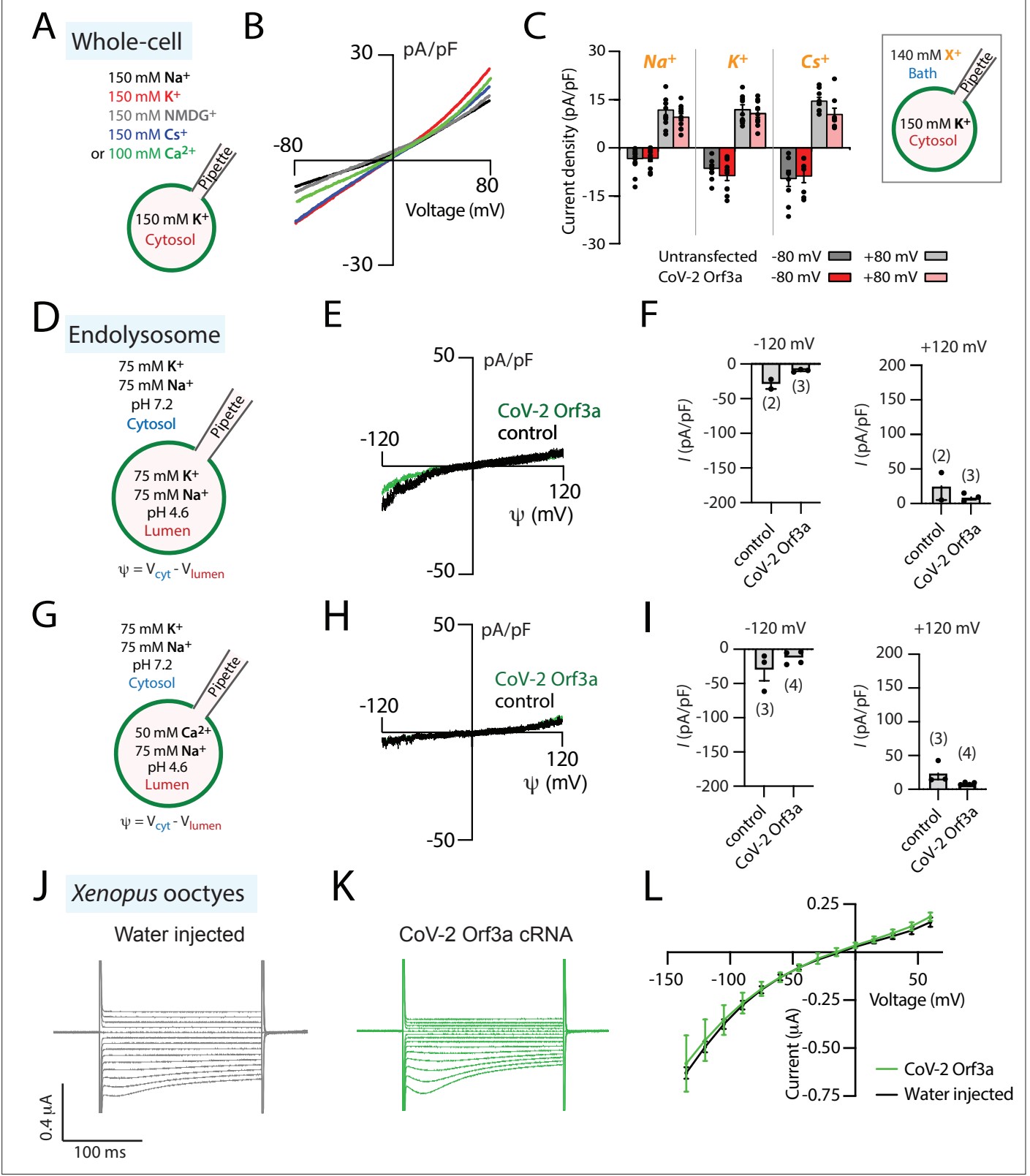

**Figure 2.** SARS-CoV-2 Orf3a is not a viroporin. (**A–C**) SARS-CoV-2 (CoV-2) Orf3a does not elicit a cation current at the plasma membrane. (**A**) Solutions used for whole-cell patch-clamp experiments. (**B**) I-V relationship for HEK293 cells expressing CoV-2 Orf3a$_{SNAP}$ by doxycycline induction in various external cationic solutions (Na$^+$, n=26; K$^+$, n=5; Cs$^+$, n=8; NMDG$^+$, n=8; Ca$^{2+}$, n=5). Mean traces are colored based on *Figure 2A*. (**C**) Average current density for untransfected HEK293 cells (gray bars) and cells transfected with CoV-2 Orf3a$_{SNAP}$ (red bars) at –80 and +80 mV recorded in Na$^+$ (n=11), K$^+$

*Figure 2 continued on next page*

*Figure 2 continued*

(n=8), and Cs⁺ (n=8) solutions. (**D–I**) CoV-2 Orf3a does not elicit a Na⁺, K⁺, or Ca²⁺-selective current in endolysosomes. (**D, G**) Solutions used in the endolysosomal patch-clamp experiments. All the bath solutions contained 150 mM Cl⁻ and pipette solutions contained 5 mM Cl⁻ (**E, H**) I-V relationship for endolysosomes from HEK293 cells expressing GFP (control, black) or CoV-2 Orf3a$_{HALO}$ (green). (**F, I**) Average current density for control and CoV-2 Orf3a$_{HALO}$ expressing HEK293 cells at –120 mV and +120 mV from (**D, G**). (**J–L**) CoV-2 Orf3a does not elicit a current in *Xenopus* oocytes when recorded in high K⁺ external solution. (**J–K**) Representative current traces from *Xenopus* oocytes injected with (**J**) water or (**K**) CoV-2 Orf3a$_{2x-STREP}$ cRNA (20 ng). Recordings are done in high external K⁺ (96 mM KCl) that reproduces published methods. (**L**) I-V relationship for water-injected (black, n=7) or CoV-2 Orf3a (green, n=7) following protocol described in (**J–K**).

The online version of this article includes the following source data and figure supplement(s) for figure 2:

**Figure supplement 1.** SARS-CoV-2 Orf3a does not elicit a H⁺-selective current in endolysosomes.

**Figure supplement 2.** SARS-CoV-2 Orf3a does not elicit a cation selective current at the plasma membrane of A549 lung alveolar cells.

**Figure supplement 3.** SARS-CoV-1 Orf3a is not a cationic ion channel at the plasma membrane of HEK293 cells and *Xenopus* oocytes.

**Figure supplement 3—source data 1.** | Raw unedited western blots and figures with the uncropped blots for *Figure 2—figure supplement 3A*.

PM localization of SARS-CoV-2 Orf3a$_{2x-STREP}$ in *Xenopus* oocytes by surface biotinylation (*Figure 2—figure supplement 3A*). These data further support the conclusion that SARS-CoV-2 Orf3a is not a viroporin.

## SARS-CoV-1 Orf3a currents are not observed in *Xenopus* oocytes or HEK293 cells

The lack of channel activity observed for SARS-CoV-2 Orf3a may reflect a functional and evolutionary distinction between SARS-CoV-2 and SARS-CoV-1 Orf3a. To explore this further, we attempted to record currents of SARS-CoV-1 Orf3a from *Xenopus* oocytes and HEK293 cells. Although we confirmed SARS-CoV-1 Orf3a PM localization in both expression systems, we recorded no currents attributable to the expressed protein (*Figure 1—figure supplement 2*, *Figure 2—figure supplement 3*; *Lu et al., 2006*; *Chan et al., 2009*; *Schwarz et al., 2011*). Our collective electrophysiology data indicate that neither SARS-CoV-1 nor SARS-CoV-2 Orf3a are viroporins.

## Overall three-dimensional architecture of SARS-CoV-2 Orf3a

Although our electrophysiological data suggest that SARS-CoV-2 Orf3a does not function as a viroporin at the PM or in the endocytic pathway, we sought to explore this further by determining two cryo-EM structures of SARS-CoV-2 Orf3a in nanodiscs that contain lipids which mimic each of these compartments (*Figure 3*, *Figure 3—figure supplements 1–4*). We first evaluated the oligomeric state of SARS-CoV-2 Orf3a$_{2x-STREP}$ in cell membranes by chemical crosslinking and concluded that it likely assembles as a 64 kDa dimer (*Figure 3—figure supplement 3I*). Despite its challenging size for cryo-EM structural determination (<100 kDa), we were able to resolve a nearly identical conformation of SARS-CoV-2 Orf3a in both nanodisc preparations, determined to 3.0 Å in the LE/Lyso environment or 3.4 Å in the PM environment (global RMSD 0.34 Å, *Figure 3A*, *Figure 3—figure supplements 1–5*, *Supplementary file 1*, *Nygaard et al., 2020*). Cryo-EM density for SARS-CoV-2 Orf3a is well-resolved between serine 40 and valine 237, whereas the electron density for the distal N- and C-termini, and a loop within the C-terminus (residues 175–180), is poor or not present. These regions are excluded from the final model (*Figure 3A*, *Figure 3—figure supplements 1–5*).

The overall architecture of SARS-CoV-2 Orf3a is a dimer and is comprised of six transmembrane (TM) helices, three provided by each subunit, with dimensions of 50 Å in diameter and 70 Å in height (*Figure 3B*). Due to the odd number of TM helices per subunit, the SARS-CoV-2 Orf3a N-terminus is positioned towards the extracellular or luminal space and its C-terminus within the cytosol, as previously reported for SARS-CoV-1 Orf3a (*Figure 3B–C*; *Lu et al., 2006*; *Tan et al., 2004*). When viewed from the extracellular/luminal side, TM1–3 are arranged in clockwise manner with pronounced tilting of TM2 and 3 (>30°) from a line perpendicular to the membrane (*Figure 3B*). Helical tilting is likely required for TM2 and 3, which are 45 Å in length, hydrophobic, and would otherwise unfavorably protrude from the membrane. TM1 is shorter by comparison (30 Å) and extends through the remainder of the lipid bilayer by a structured loop that connects with TM2. Although tilted at a similar angle in the membrane, TM2 and 3 are positioned at a 30° angle with respect to one another (*Figure 3B*). Consequently, the combined angle between TM2-3 and the structured loop between TM1-2 create

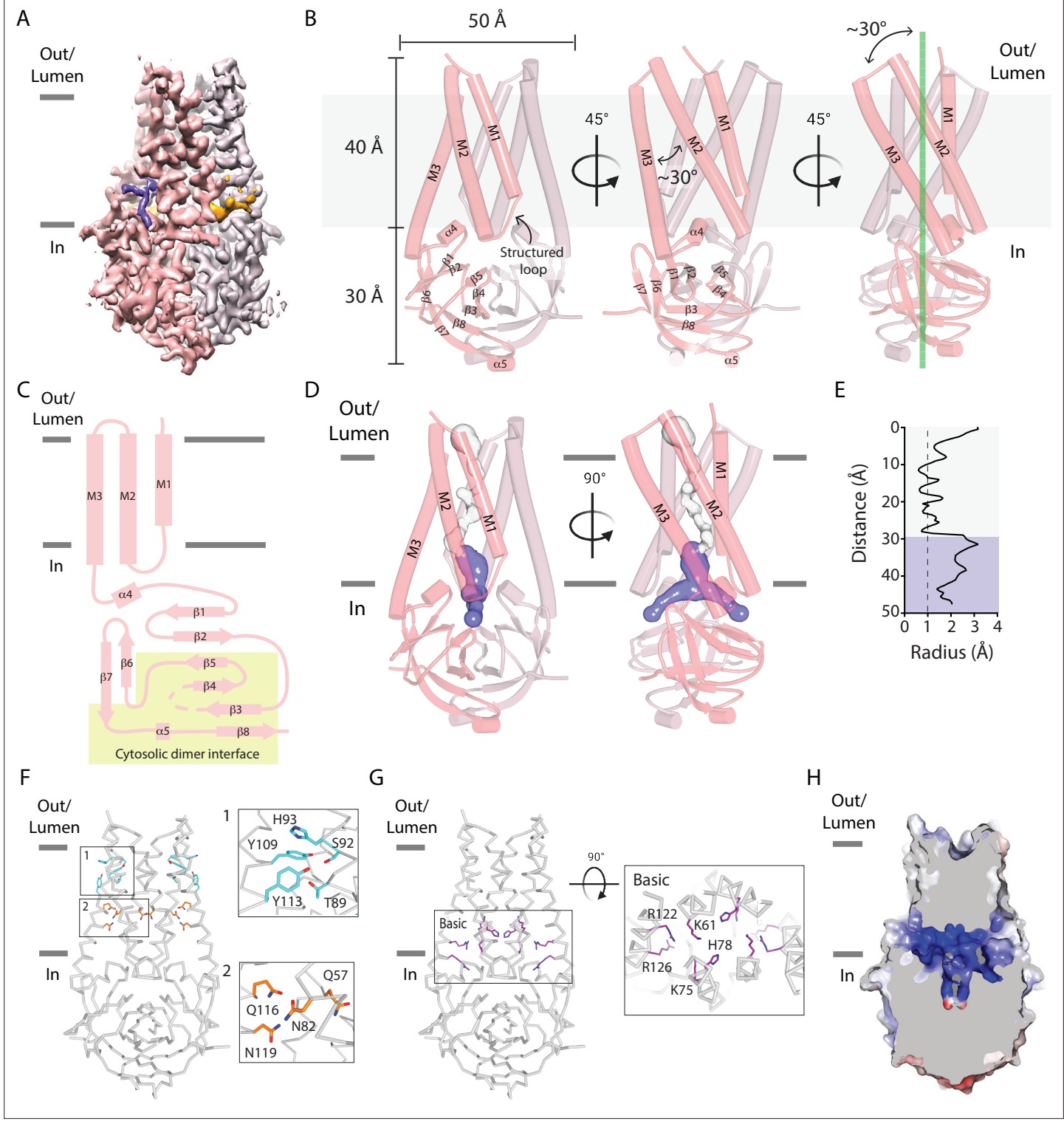

**Figure 3.** A narrow cavity detected in the SARS-CoV-2 Orf3a TM region is unlikely to conduct cations. (**A–C**) Overall architecture of SARS-CoV-2 (CoV-2) Orf3a. (**A**) Cryo-EM map of dimeric CoV-2 Orf3a (dark and light pink), with density for lipids colored (orange, purple). (**B**) Three side views of CoV-2 Orf3a depicting dimeric architecture (dark and light pink) and key structural elements. (**C**) 2D topology of CoV-2 Orf3a. The region forming the cytosolic dimer interface is shown (yellow). (**D**) Inspection of the CoV-2 Orf3a TM region for a pore, depicted as the minimal radial distance from its center to the nearest van der Waals contact (HOLE program) (**Smart et al., 1996**). A region too narrow to conduct ions (white) and an aqueous vestibule (dark blue) are highlighted. (**E**) Radius of the pore (from **D**) as a function of the distance along the ion pathway. Dashed lines indicate the minimal radius that would permit a dehydrated ion. Blue and white colors follow (**D**). (**F**) Two layers of polar residues (1 and 2, cyan and orange) identified in the TM region,

*Figure 3 continued on next page*

*Figure 3 continued*

with a zoom in of each region. (**G**) Basic residues located in the aqueous vestibule (purple) with zoom in of the region. (**H**) Cutaway of the CoV-2 Orf3a molecular surface to view the aqueous vestibule is colored according to the electrostatic potential (APBS program) (*Jurrus et al., 2018*). Coloring: blue, positive (+10 kT/e) and red, negative (–10 kT/e).

The online version of this article includes the following figure supplement(s) for figure 3:

**Figure supplement 1.** Cryo-EM data processing workflow for SARS-CoV-2 Orf3a reconstituted in LE/Lysosomal MSP1D1-containing nanodiscs.

**Figure supplement 2.** Cryo-EM data processing workflow for SARS-CoV-2 Orf3a reconstituted in LE/Lysosomal MSP1D1-containing nanodiscs, continued from *Figure 3—figure supplement 1*.

**Figure supplement 3.** Structural determination of SARS-CoV-2 Orf3a LE/Lyso (**A–D**) or PM (**E–H**) MSP1D1 nanodiscs.

**Figure supplement 4.** Cryo-EM data processing workflow for SARS-CoV-2 Orf3a reconstituted in PM MSP1D1-containing nanodiscs.

**Figure supplement 5.** Representative cryo-EM density for SARS-CoV-2 Orf3a and SARS-CoV-1 Orf3a structures.

several gaps per subunit that expose the protein core to the membrane, accommodating weak to moderately resolved electron density which we attribute to lipids (*Figure 3A, B* and *Figure 4A, B*, *Figure 4—figure supplement 1*).

Extending from TM3 is a 104 amino acid structured cytosolic domain assembled from a cluster of eight β-sheets (β1–8) contributed by each SARS-CoV-2 Orf3a subunit, forming a compact and continuous molecular surface that protrudes 30 Å into the cytosol (*Figure 3B–C*). Packing of the dimer is facilitated by β3–5 and β8, as well as loops joining β2–3, β4–5 and a helical turn and loop connecting β7–8 at its base (*Figure 3B–C*). These extensive interactions along the dimer interface within the cytosolic domain (buried surface interface of 1010 Å² per subunit) appear integral to the integrity of the SARS-CoV-2 Orf3a dimer (*Figure 3C*).

## A narrow cavity detected in the SARS-CoV-2 Orf3a transmembrane region likely does not represent a pore

The hallmark of all ion channels is an aqueous and often hydrophilic pore that spans the TM region of the protein. To structurally evaluate whether SARS-CoV-2 Orf3a may be a viroporin that conducts $K^+$ or other cations, we inspected the TM region of the protein for a channel pore that fits these criteria. Approximately two-thirds of the TM region is tightly packed in its current conformational state, with the narrowest point in this region (~0.8 Å radius) too narrow to accommodate a dehydrated cation (*Figure 3D–E*). We then inspected the amino acid composition within this region to identify polar or charged residues which might form a hydrophilic pore if SARS-CoV-2 Orf3a was in a different conformational state. Two distinct clusters of polar residues, the first positioned towards the extracellular or luminal space (T89, S92, H93, Y109, Y113) and the second situated within the center of the membrane (Q57, N82, Q116, N119) are regions that may accommodate hydrophilic ions or molecules (*Figure 3F*).

The final approximately one-third of the TM region positioned above the cytosolic domain contains a ~12 Å diameter aqueous vestibule which is accessible from the cytosol through two narrow portals. Although the portal could permit the movement of partially hydrated cations (1.5 Å radius), the composition of residues lining the aqueous vestibule is highly basic (K61, K75, H78, R122, R126) creating a positively-charged region that would not be suitable for cations (*Figure 3D–E and G–H*). The lack of a clear and identifiable pore is inconsistent with an ion channel, even if captured in a closed or non-conductive state.

## Two lateral openings within the transmembrane region are filled with electron densities attributable to lipids

Within the SARS-CoV-2 Orf3a transmembrane region are two distinct lateral fenestrations that expose the aqueous vestibule to the lipid bilayer formed between the structured loop of TM1 and TM3 (Lipid Site 1) and between TM2 and TM3 (Lipid Site 2) (*Figure 4A–C*, *Figure 4—figure supplement 1*). Each fenestration is filled with tubular density attributable to lipids (*Figure 4—figure supplement 1*). We modeled a 1,2-Dioleoyl-sn-glycero-3-phosphatidylethanolamine (DOPE) molecule into Lipid Site 1 based on the size, shape, and presence of this density in both PM and LE/Lyso lipid compositions (*Figure 4B*, *Figure 4—figure supplement 1*). Similarly, we assigned DOPE to Lipid Site 2 (*Figure 4C*, *Figure 4—figure supplement 1*). A notable arginine residue (R122) located in TM3 neighbors both

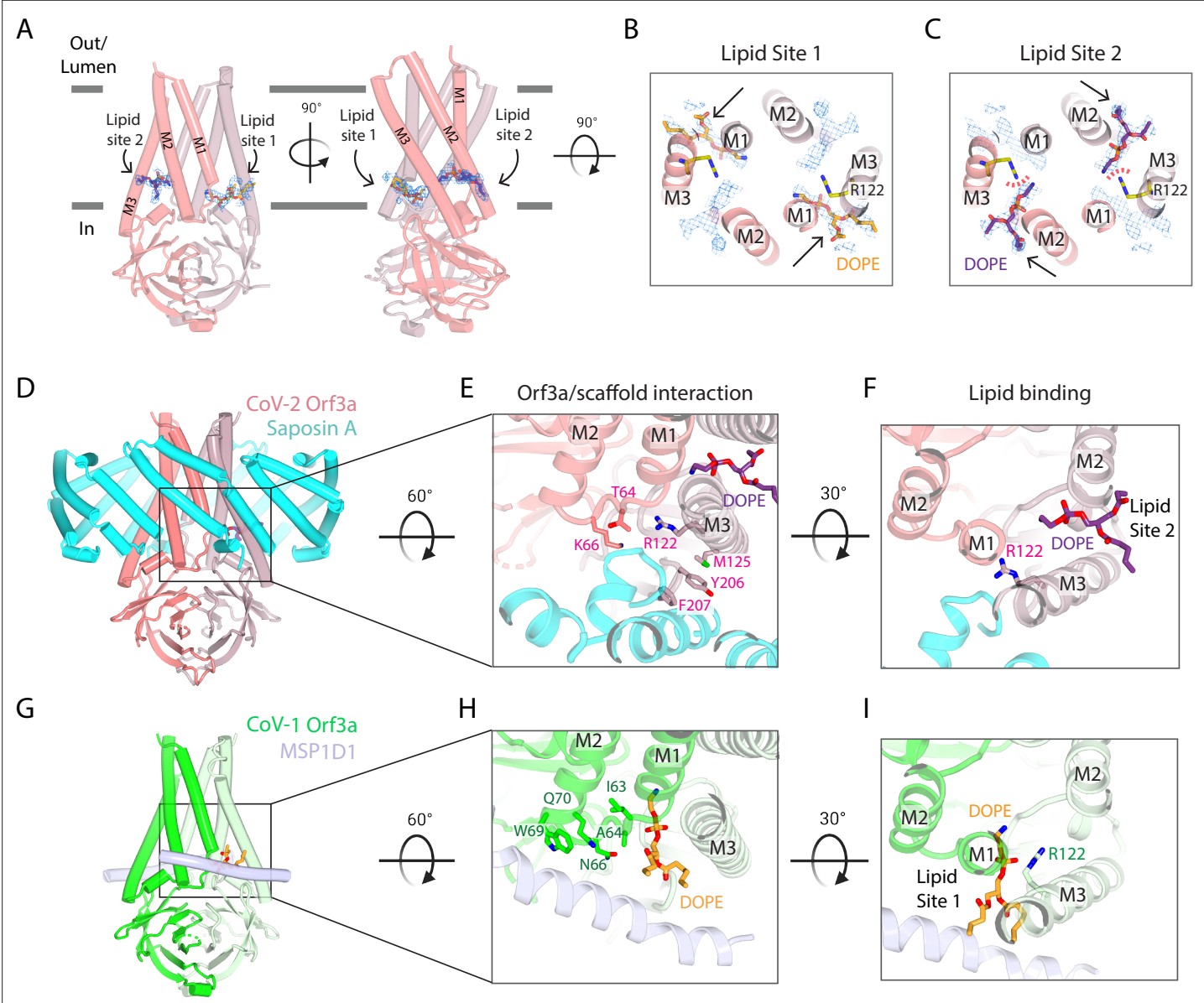

**Figure 4.** Two SARS-CoV-2 Orf3a lateral openings within the TM region are filled with density likely representing lipid sites. (**A**) Two side views of SARS-CoV-2 (CoV-2) Orf3a in LE/Lyso MSP1D1 nanodiscs highlighting two subunits (dark and light pink). Lipid densities (blue mesh, contoured at 7σ) are identified in fenestrations between TM1 and TM3 of neighboring subunits (Lipid Site 1) and TM2 and TM3 of the same subunit (Lipid Site 2). A DOPE lipid is modeled into Lipid Site 1 (orange) and Lipid Site 2 (purple) in each view. (**B–C**) Cutaway view from the extracellular space to view lipids modeled into the density. Two DOPE lipids are modeled into (**B**) Lipid Site 1 (orange) or (**C**) Lipid Site 2 (purple) with lipid density depicted (blue mesh, contoured at 7σ). Lipid Sites 1 and 2 are likely not occupied simultaneously since the orientation of R122 (yellow) would sterically clash with DOPE in Lipid Site 2 (red dotted line). (**D–F**) CoV-2 Orf3a interacts with Saposin A. (**D**) Side view of CoV-2 Orf3a in LE/Lyso Saposin A nanodiscs highlighting two subunits (dark and light pink) and 6 molecules of Saposin A (cyan), with DOPE shown (purple). (**E**) Zoom in from the extracellular side to highlight the CoV-2 Orf3a and Saposin A interaction. CoV-2 Orf3a residues within 5 Å from Saposin A are shown (light pink) with DOPE (purple). (**F**) Zoom in from the extracellular side to highlight DOPE in Lipid Site 2 (purple). Note that residue R122 (light pink) is rotated 135° from the CoV-2 Orf3a LE/Lyso MSP1D1 structure (compare with *Figure 4B*; see also *Figure 4I* for direct comparison) and occludes Lipid Site 1. (**G–I**) SARS-CoV-1 (CoV-1) Orf3a interacts with MSP1D1. (**G**) Side view of CoV-1 Orf3a in LE/Lyso MSP1D1 nanodiscs highlighting two subunits (dark and light green) and two molecules of MSP1D1 (light blue), with DOPE shown (orange). (**H**) Same view as *Figure 4E* to highlight the CoV-1 Orf3a and MSP1D1 interaction. CoV-1 Orf3a residues within 5 Å of MSP1D1 are shown (green), with DOPE (orange) depicted. (**I**) View as in *Figure 4F* to highlight DOPE in Lipid Site 1 (orange). Similar to *Figure 4B-C*, residue R122 (green) is positioned near and clashes with Lipid Site 2.

The online version of this article includes the following figure supplement(s) for figure 4:

*Figure 4 continued on next page*

*Figure 4 continued*

**Figure supplement 1.** Comparison of lipid densities between (**A, D**) SARS-CoV2 Orf3a LE/Lyso MSP1D1-containing nanodiscs, (**B, E**) SARS-CoV2 Orf3a LE/Lyso Saposin A-containing nanodiscs, and (**C, F**) SARS-CoV-1 Orf3a LE/Lyso MSP1D1-containing nanodiscs.

**Figure supplement 2.** Cryo-EM data processing workflow for SARS-CoV-2 Orf3a reconstituted in LE/Lyso Saposin A-containing nanodiscs.

**Figure supplement 3.** Structural determination of SARS-CoV-2 Orf3a LE/Lyso Saposin A nanodisc (**A–D**) or SARS-CoV-1 LE/Lyso MSP1D1 nanodisc (**E–H**).

**Figure supplement 4.** Cryo-EM data processing workflow for SARS-CoV-1 Orf3a reconstituted in LE/Lysosomal MSP1D1-containing nanodiscs.

**Figure supplement 5.** A similar narrow cavity is detected in the TM region of SARS-CoV-1 Orf3a.

lipid sites and, in its current conformation, stabilizes the phospholipid phosphate group in Lipid Site 1 (*Figure 4B*). It is unlikely that both fenestrations would be occupied by DOPE at once since their positively-charged headgroups are in proximity (~5 Å) to one another and R122 clashes with Lipid Site 2 (*Figure 4C*). DOPE bound in either lipid site would contribute to the positive electrostatic landscape of the aqueous vestibule.

## Structure of SARS-CoV-2 Orf3a in a Saposin A containing nanodisc provides additional insight into lipid fenestration binding

In both SARS-CoV-2 Orf3a cryo-EM maps, we observed low-resolution density for the membrane scaffold protein (MSP) used for nanodisc assembly (MSP1D1, *Figure 3—figure supplement 4*). MSPs typically assemble as two belts that wrap around the lipid bilayer, shielding its hydrophobic edges from the aqueous environment. However, we observe direct binding of the MSP1D1 with SARS-CoV-2 Orf3a, which is unusual and may inadvertently stabilize a single conformational state of the protein (*Figure 3—figure supplement 4*). To circumvent this and potentially capture a different conformational state, we substituted MSP1D1 with another scaffold protein, Saposin A (*Figure 4—figure supplements 2–3*; *Popovic et al., 2012*; *Frauenfeld et al., 2016*; *Flayhan et al., 2018*; *Nguyen et al., 2018*). We determined the cryo-EM structure of SARS-CoV-2 Orf3a in a LE/Lyso lipid Saposin A-containing nanodisc to 2.8 Å resolution (*Figure 4—figure supplement 3*, *Supplementary file 1*). Its overall architecture and conformational state are nearly identical to the structures of SARS-CoV-2 Orf3a in MSP1D1-containing nanodiscs, harboring a TM constriction and a basic aqueous vestibule (global RMSD 0.50 Å, *Figure 4—figure supplement 3I*). We observe a direct interaction of Saposin A with SARS-CoV-2 Orf3a, which occurs in a similar region to where MSP1D1 is binding, but with a slightly different interface of SARS-CoV-2 Orf3a (*Figure 4D*, *Figure 4—figure supplement 2*). Although we were unable to identify another conformational state from this dataset, the discrete interfaces of interaction between the two scaffold proteins and SARS-CoV-2 Orf3a, with little change to the SARS-CoV-2 Orf3a structure, increases our confidence that our cryo-EM structures represent a native conformational state of the protein.

Comparison of the SARS-CoV-2 Orf3a MSP1D1 and Saposin A nanodisc structures reveals several differences which can be attributed to scaffold protein binding. The binding of the Saposin A to SARS-CoV-2 Orf3a directly occludes Lipid Site 1 and consequently, electron density is absent from this site. Instead, we observe a 135° rotation of R122 into Lipid Site 1, where its side chain is directly interacting with Saposin A, creating space for a lipid to occupy Lipid Site 2 (*Figure 4F*, *Figure 4—figure supplement 1*). The rotation of R122 side chain and the electron density exclusively in Lipid Site 2 supports the argument that Lipid Sites 1 and 2 are not simultaneously occupied. The presence of density in both Lipid Sites 1 and 2 in the MSP1D1-containing SARS-CoV-2 Orf3a cryo-EM maps likely represents two lipid bound states that are averaged together during the 3D reconstruction.

## The three-dimensional architecture of SARS-CoV-1 Orf3a is nearly identical to SARS-CoV-2 Orf3a

To further address the possibility of a functional and evolutionary distinction between SARS-CoV-2 and SARS-CoV-1 Orf3a, we determined the structure of SARS-CoV-1 Orf3a in a LE/Lyso-like membrane MSP1D1 nanodisc to 3.1 Å resolution and compared it to the SARS-CoV-2 homolog (*Figure 4—figure supplements 3–5*, *Supplementary file 1*). Its overall architecture and conformational state are nearly identical to the SARS-CoV-2 Orf3a cryo-EM structures (global RMSD 0.47 Å, *Figure 4—figure supplement 3J*). SARS-CoV-1 Orf3a does not have an obvious pore in its current conformational state, and its

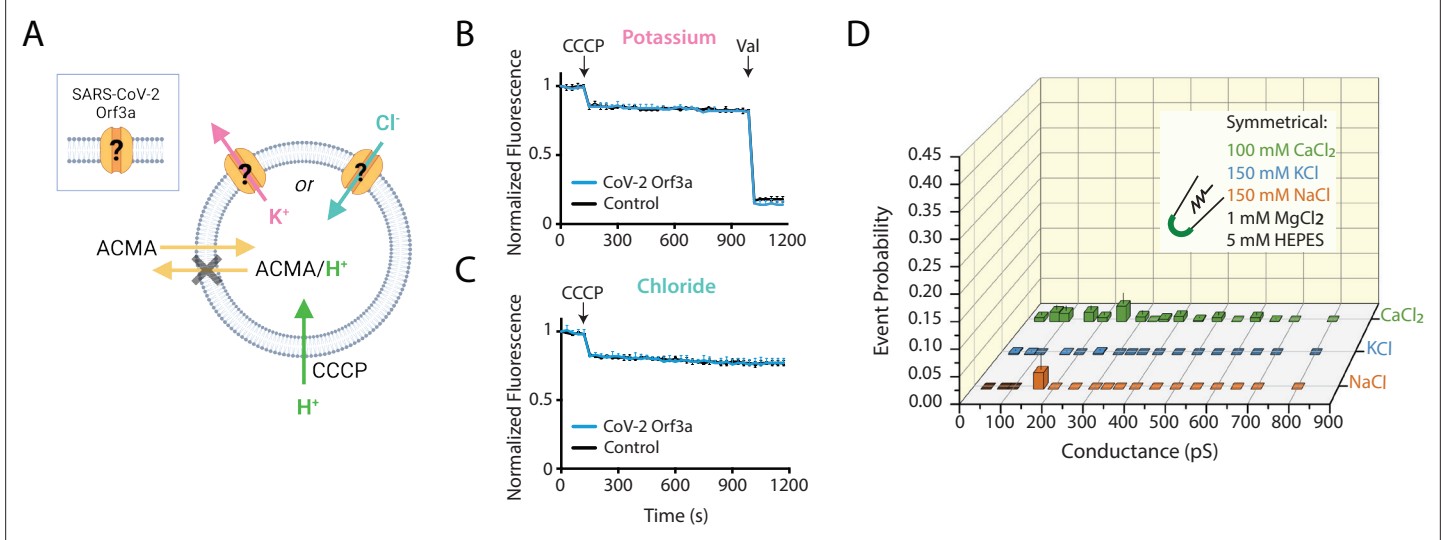

**Figure 5.** SARS-CoV-2 Orf3a does not elicit ion flux or conductances in a vesicle-reconstituted system. (**A**) Schematic of the ACMA-based fluorescence flux assay (***Zhang et al., 1994***; ***Heginbotham et al., 1998***; ***Miller and Long, 2012***; ***Kane Dickson et al., 2014***). A K$^+$ (pink) or Cl$^-$ (blue) gradient is generated by reconstitution and dilution into an appropriate external salt solution (K$^+$ efflux: 150 KCl in, 150 NMDG-Cl out; Cl$^-$ flux: 110 Na$_2$SO$_4$ in, 125 NaCl out; in mM). If CoV-2 Orf3a conducts K$^+$ or Cl$^-$ ions, then the addition of the protonophore carbonyl cyanide m-chlorophenyl hydrazone (CCCP) will drive H$^+$ (green) influx. ACMA is quenched and sequestered in vesicles at low pH, resulting in loss of ACMA fluorescence. Valinomycin (Val), a K$^+$ permeable ionophore, is added to the end of the K$^+$ flux assay to empty all vesicles. Created with Biorender.com. (**B–C**) K$^+$ (n=4) (**B**) or Cl$^-$ (n=4) (**C**) flux is not observed in SARS-CoV-2 (CoV-2) Orf3a$_{2x\text{-STREP}}$-reconstituted vesicles (blue) as compared with the empty vesicle control (black, n=4) using vesicles reconstituted at a 1:100 (wt:wt) protein to lipid ratio. CCCP and Val are added as indicated (arrows). Error is represented as SEM. (**D**) Probability of observing an open event in a CoV-2 Orf3a$_{2x\text{-STREP}}$-reconstituted proteoliposome patch with vesicle reconstituted at a 1:100 protein to lipid ratio. NaCl, n=27; KCl, n=32, and CaCl$_2$, n=105. Error is represented as SEM.

The online version of this article includes the following figure supplement(s) for figure 5:

**Figure supplement 1.** Characterization of vesicle-reconstituted SARS-CoV-1 and SARS-CoV-2 Orf3a at low and high protein ratios.

**Figure supplement 2.** Multiple conductance species are observed from SARS-CoV-2 Orf3a containing-vesicles reconstituted at a high protein to lipid ratio and likely result from transient membrane leakiness and/or contamination by *bona fide* ion channels.

aqueous vestibule is also electrostatically positive (***Figure 4—figure supplement 5***). In combination with our extensive efforts to identify and reproduce SARS-CoV-1 and SARS-CoV-2 Orf3a currents in various expression systems without success, we conclude that both Orf3a homologs are not viroporins.

Further supporting the idea of two distinct lipid bound states, we observe pronounced electron density for the MSP1D1 scaffold protein in this dataset and built a model (***Figure 4—figure supplement 3***). MSP1D1 binds to SARS-CoV-1 Orf3a in a similar area as Saposin A (***Figure 4E***, ***Figure 4—figure supplement 3***). Accordingly, density was observed in Lipid Site 1 with weak density observed in Lipid Site 2, suggesting a preferred Lipid Site 1 bound state in maps of Orf3a that contain resolved MSP1D1 density (***Figure 4E and G***, ***Figure 4—figure supplement 1***). This difference between the SARS-CoV-1 and SARS-CoV-2 Orf3a MSP1D1-containing nanodisc structures likely reflects a variation in particle heterogeneity between the datasets and not a distinction between Orf3a proteins. A recently published cryo-EM map of SARS-CoV-2 Orf3a in a PM lipid MSP1D1-containing nanodisc resolved a high-resolution density for MSP1D1 and a concomitant Lipid Site 1-only bound state (***Kern et al., 2021***). It is unclear what the significance of the discrete lipid bound states may be, but may suggest a function of Orf3a proteins that involves interaction with lipids in the bilayer.

## Macroscopic K$^+$ and Cl$^-$ flux is not observed in vesicle-reconstituted SARS-CoV-1 and SARS-CoV-2 Orf3a

To further evaluate whether SARS-CoV-2 Orf3a might form a viroporin, we performed both flux assays and proteoliposome patch-clamp experiments with purified, vesicle-reconstituted SARS-CoV-2 Orf3a (***Figure 5***; ***Kern et al., 2021***; ***Miller, 1986***). We reconstituted purified SARS-CoV-2 Orf3a$_{2x\text{-STREP}}$ at a high (1:10, 1:25) or standard (1:100, wt:wt) protein to lipid ratio to ensure detection of transport and

ionic currents, and to attempt to recapitulate published results (*Kern et al., 2021*). We did not observe any K$^+$ or Cl$^-$ transport using a 9-Amino-6-Chloro-2-Methoxyacridine (ACMA) fluorescence-based flux assay (*Figure 5B–C*, *Figure 5—figure supplement 1A-C*). However, the counter ions used in these assays could also permeate through a non-selective viroporin, eliminating the ion gradient needed to generate flux. To circumvent this, we designed a 90° light-scattering K$^+$ flux assay (*Figure 5— figure supplement 1D*; *Brammer et al., 2014*; *Stockbridge et al., 2012*). Compared to the vesicle control, no reduction in light scattering is observed with SARS-CoV-2 Orf3a$_{2x-STREP}$-containing vesicles (*Figure 5—figure supplement 1E–G*).

## Large single-channel currents measured from vesicle-reconstituted SARS-CoV-2 Orf3a are due to transient membrane leakiness and/or channel contamination

We next asked if we could record macroscopic channel currents by proteoliposome patch-clamp using vesicle-reconstituted SARS-CoV-2 Orf3a. Our attempts to record macroscopic currents were unsuccessful, yet we observed large single channel-like conductances in symmetrical K$^+$, Na$^+$ and Ca$^{2+}$ from SARS-CoV-2 Orf3a$_{2x-STREP}$ vesicles reconstituted at a high, but not standard, protein to lipid ratio (*Figure 5D*, *Figure 5—figure supplement 2A–C*). Although these conductances are consistent to what has been recently published, we believe that these data likely result from transient membrane leakiness or ion channel protein contamination, and not due to SARS-CoV-2 Orf3a activity (*Figure 5—figure supplement 2B–C*; *Kern et al., 2021*; *Accardi et al., 2004*; *Niu et al., 2021*). We examined the possibility that the single channel current measured could be contributed by an ion channel that was inadvertently co-purified with SARS-CoV-2 Orf3a. We performed mass spectrometry using proteoliposome-reconstituted SARS-CoV-2 Orf3a. The only ion channel proteins that appeared on this list were the voltage-dependent anion channels 1–2 (VDAC1–2), outer membrane mitochondrial proteins that are large-conducting, non-selective channels which can be blocked by 4,4'-Diisothiocyano-2,2'-stilbenedisulfonic Acid (DIDS) (*Figure 5—figure supplement 2D–F*; *Thinnes et al., 1994*). We asked if some of the currents that we record could be blocked by DIDS. After 1 min of application of the vehicle (DMSO), we observed channel block by 100 µM DIDS, which could be recovered by a washout of the inhibitor, suggesting that some of the current that we measure may be contributed by a VDAC (*Figure 5—figure supplement 2E–F*). This was a reminder that patch-clamp recording detects single proteins (as single channel openings) and thus far exceeds the limits of specific protein biochemical purification (*Accardi et al., 2004*). This phenomenon has been described for samples reconstituted at a high protein to lipid ratio, and is a common cause of novel protein misidentification as ion channels (*Niu et al., 2021*; *Miller, 1986*).

## SARS-CoV-2 Orf3a interacts with VPS39, a member of the HOPS complex, altering fusion of late endosomes with lysosomes

If SARS-CoV-2 Orf3a is not a viroporin, then what is its function? We turned to published proteomics datasets that have identified putative host proteins which interact with SARS-CoV-2 Orf3a to glean insight (*Gordon et al., 2020a*; *Gordon et al., 2020b*). One of these host proteins, VPS39, is a member of the HOPS complex, a membrane tethering complex that facilitates LE or autophagosome (AP) fusion with lysosomes (*Balderhaar and Ungermann, 2013*). Defects in HOPS complex-mediated trafficking leads to the accumulation of these organelles (*Liang et al., 2008*; *Pols et al., 2013*; *Jiang et al., 2014*; *Takáts et al., 2014*; *Aoyama et al., 2012*; *Messler et al., 2011*). In doxycycline-treated HEK293 cells expressing SARS-CoV-2 Orf3a$_{HALO}$, we observe an accumulation of Rab7 puncta, which is not observed in uninduced, control cells, suggesting a defect in HOPS complex-mediated vesicle fusion (*Figure 6A and J*; *Miao et al., 2021*; *Chen et al., 2021*; *Qu et al., 2021*). We then asked if a similar phenomenon is observed in the presence of SARS-CoV-1 Orf3a. In contrast, we do not observe the Rab7 puncta after doxycycline-induced expression of SARS-CoV-1 Orf3a$_{HALO}$, suggesting an evolutionary distinction between these two homologs (*Figure 6B and K*; *Miao et al., 2021*; *Chen et al., 2021*; *Qu et al., 2021*).

We next asked whether SARS-CoV-1 and SARS-CoV-2 Orf3a bind to the HOPS complex protein, VPS39. We anticipated that VPS39 likely interacts with SARS-CoV-2 Orf3a, but not SARS-CoV-1 Orf3a, based on our Rab7 accumulation data. For these co-immunoprecipitation (co-IP) experiments, we titrated increasing concentrations of purified Orf3a$_{2x-STREP}$ with cell lysate containing overexpressed

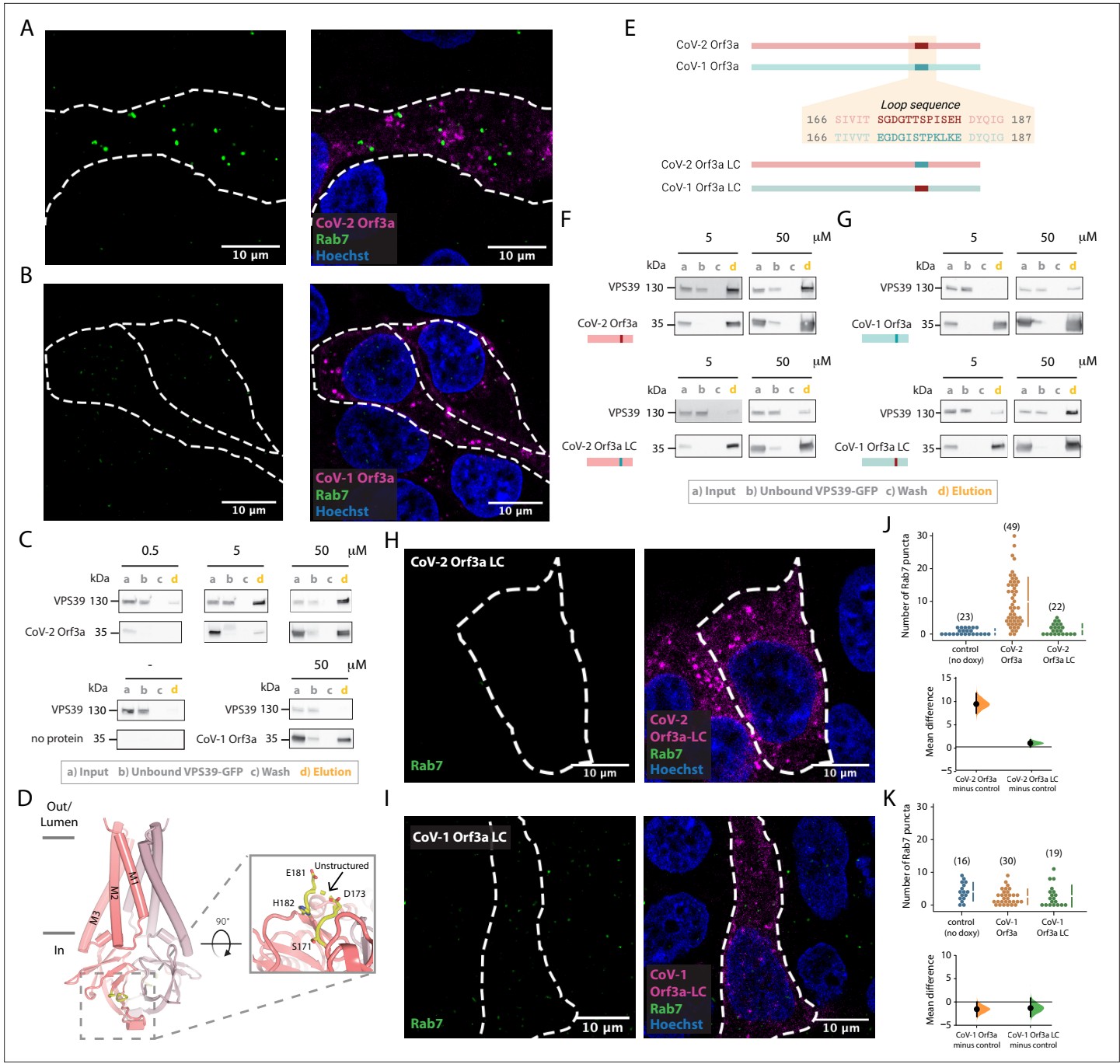

**Figure 6.** SARS-CoV-2 Orf3a, but not SARS-CoV-1 Orf3a, interacts with HOPS protein, VPS39. (**A–B**) Rab7 puncta (green) are abundant in HEK293 cells expressing (**A**) SARS-CoV-2 (CoV-2) Orf3a$_{HALO}$ (magenta), but not (**B**) SARS-CoV-1 (CoV-1) Orf3a$_{HALO}$ (magenta; Hoechst 33342, blue). (**C**) Co-immunoprecipitation (co-IP) evaluating the interaction of VPS39$_{GFP}$ with CoV-1 and CoV-2 Orf3a$_{2x-STREP}$, detected by western blot with antibodies against GFP and streptavidin, respectively. VPS39$_{GFP}$ elutes with CoV-2 Orf3a$_{2x-STREP}$ in a concentration-dependent manner, but does not elute with purified CoV-1 Orf3a$_{2x-STREP}$ (compare VPS39 in d lanes, orange). Control, co-IP without Orf3a$_{2x-STREP}$ added (bottom left, no protein). VPS39$_{GFP}$ and Orf3a$_{2x-STREP}$ migrate at ~130 and 35 kDa, respectively, by SDS-PAGE. (**D–I**) An unstructured loop of CoV-2 Orf3a partially mediates its interaction with VPS39. (**D**) Side view of CoV-2 Orf3a structure with the subunits (dark and light pink) and unstructured loop highlighted (yellow, dotted box). Zoom-in of the loop from the cytosol (solid box) with resolved loop residues. (**E**) CoV-2 Orf3a (red) and CoV-1 Orf3a (blue green) loop sequences. Orf3a wild-type (WT) and loop chimeras (LC) are color matched or swapped. Created with Biorender.com (**F**) Co-IP as in *Figure 6C* with CoV-2 Orf3a constructs showing loss of VPS39$_{GFP}$ elution with CoV-2 Orf3a LC$_{2x-STREP}$ (compare VPS39 in d lanes, orange). The co-IPs presented in this figure represent three to seven independent experiments. (**G**) Co-IP of VPS39$_{GFP}$ with CoV-1 Orf3a constructs shows an enrichment with CoV-1 Orf3a LC$_{2x-STREP}$. (**H, I**) Rab7 puncta (green)

*Figure 6 continued on next page*

Figure 6 continued

are absent in CoV-2 Orf3a LC$_{HALO}$ (**H**, magenta) or CoV-1 Orf3a LC$_{HALO}$-expressing HEK293 cells (**I**, magenta; Hoechst 33342, blue), consistent with **Chen et al., 2021**. (**J–K**) Cumming estimation plots of Rab7 puncta from (**A, H**) (**J**) and (**B, I**) (**K**) (**Ho et al., 2019**).

The online version of this article includes the following source data and figure supplement(s) for figure 6:

**Source data 1.** Raw unedited western blots and figures with the uncropped blots for **Figure 6C**.

**Source data 2.** Raw unedited western blots and figures with the uncropped blots for **Figure 6F**.

**Figure supplement 1.** Purification of SARS-CoV-1 and SARS CoV-2 Orf3a loop chimeras.

VPS39$_{GFP}$ (**Figure 6C**). We observed an enrichment of VPS39$_{GFP}$ that was SARS-CoV-2 Orf3a$_{2x-STREP}$ concentration dependent (0.5–50 μg protein; **Figure 6C**), but very little to no binding was observed when performing the co-IP with all concentrations of SARS-CoV-1$_{2x-STREP}$ Orf3a (50 μg shown; **Figure 6C**).

### An unstructured loop unique to SARS-CoV-2 Orf3a interacts with VPS39

Since we observe an interaction of VPS39 with SARS-CoV-2 Orf3a, but not SARS-CoV-1 Orf3a, we asked if we could identify the site of the protein-protein interaction. By inspecting the cytosolic domain of both Orf3a cryo-EM structures and by pinpointing the divergent regions of amino acid sequence conservation, we identified an unstructured loop between β3–4 that could be facilitating the VPS39 interaction (**Figure 6D**). We generated chimeras of SARS-CoV-2 and SARS-CoV-1 Orf3a by swapping the 12 amino acid loop sequence between these two proteins and observed loss of the interaction between SARS-CoV-2 Orf3a$_{2x-STREP}$ loop chimera (LC) and VPS39$_{GFP}$ (**Figure 6E and F**, **Figure 6—figure supplement 1**). Conversely, by substituting the SARS-CoV-2 Orf3a 12 amino acid loop into the SARS-CoV-1 Orf3a (SARS-CoV-1 Orf3a$_{2x-STREP}$ LC), we noticed enhanced interaction with VPS39$_{GFP}$ compared with SARS-CoV-1 Orf3a$_{2x-STREP}$ (**Figure 6E and G**, **Figure 6—figure supplement 1**). However, this substitution did not fully restore the binding of VPS39 with SARS-CoV-2 Orf3a (compare with **Figure 6C and F**).

We next asked if overexpression of SARS-CoV-1 and SARS-CoV-2 Orf3a LC could alter HOPS complex-mediated fusion, as assessed by the accumulation of Rab7-containing vesicles. Overexpression of SARS-CoV-2 Orf3a$_{HALO}$ LC by doxycycline in HEK293 cells did not promote Rab7 accumulation, consistent with its weaker interaction with VPS39 (**Figure 6F, H and J**). SARS-CoV-1 Orf3a$_{HALO}$ LC also did not promote Rab7 accumulation, as one would expect if the Orf3a and VPS39 interaction was fully rescued (**Figure 6I**). This is consistent with the co-IP experiments and suggests that the binding interface involves more than the unstructured loop (**Figure 6G1 and K**, **Figure 7A**). Overall, we observe an interaction of VPS39 that is specific to SARS-CoV-2 Orf3a, resulting in a Rab7 accumulation phenotype typical of a HOPS complex fusion defect, and that this interaction is partially mediated by a divergent, unstructured loop of SARS-CoV-2 Orf3a.

## Discussion

Viroporins are small viral membrane proteins that often form weakly- or non-selective pores (**Nieva et al., 2012**; **Scott and Griffin, 2015**). They are widely distributed among different viral families and, accordingly, their contributions to viral survival range widely from assisting with viral propagation to participating in host immune evasion (**Nieva et al., 2012**; **Scott and Griffin, 2015**). Four proteins encoded by SARS-CoV-1 have been proposed to function as viroporins - E protein, Orf3a, Orf8a and Orf10 (**Lu et al., 2006**; **Harrison et al., 2022**; **Liao et al., 2004**; **Chen et al., 2011**). To demonstrate that a novel protein unequivocally forms a viroporin requires the identification of its pore-lining residues. This is ideally done by introducing a pore-lining cysteine that can be modified by a methan-ethiosulfonate reagent, which when added, partially blocks the pore or alters ion channel selectivity (**Newell and Czajkowski, 2007**; **Frillingos et al., 1998**). This has yet to be demonstrated for SARS-CoV-1 or SARS-CoV-2 Orf3a.

We asked if Orf3a from SARS-CoV-2 is a *bona fide* viroporin and took a comprehensive approach to investigate its function in several mammalian cell lines. We first assessed subcellular localization of SARS-CoV-2 Orf3a to determine where it may be functioning in mammalian cells and observed its enrichment at the PM and in the endocytic pathway. We then performed whole-cell patch-clamp

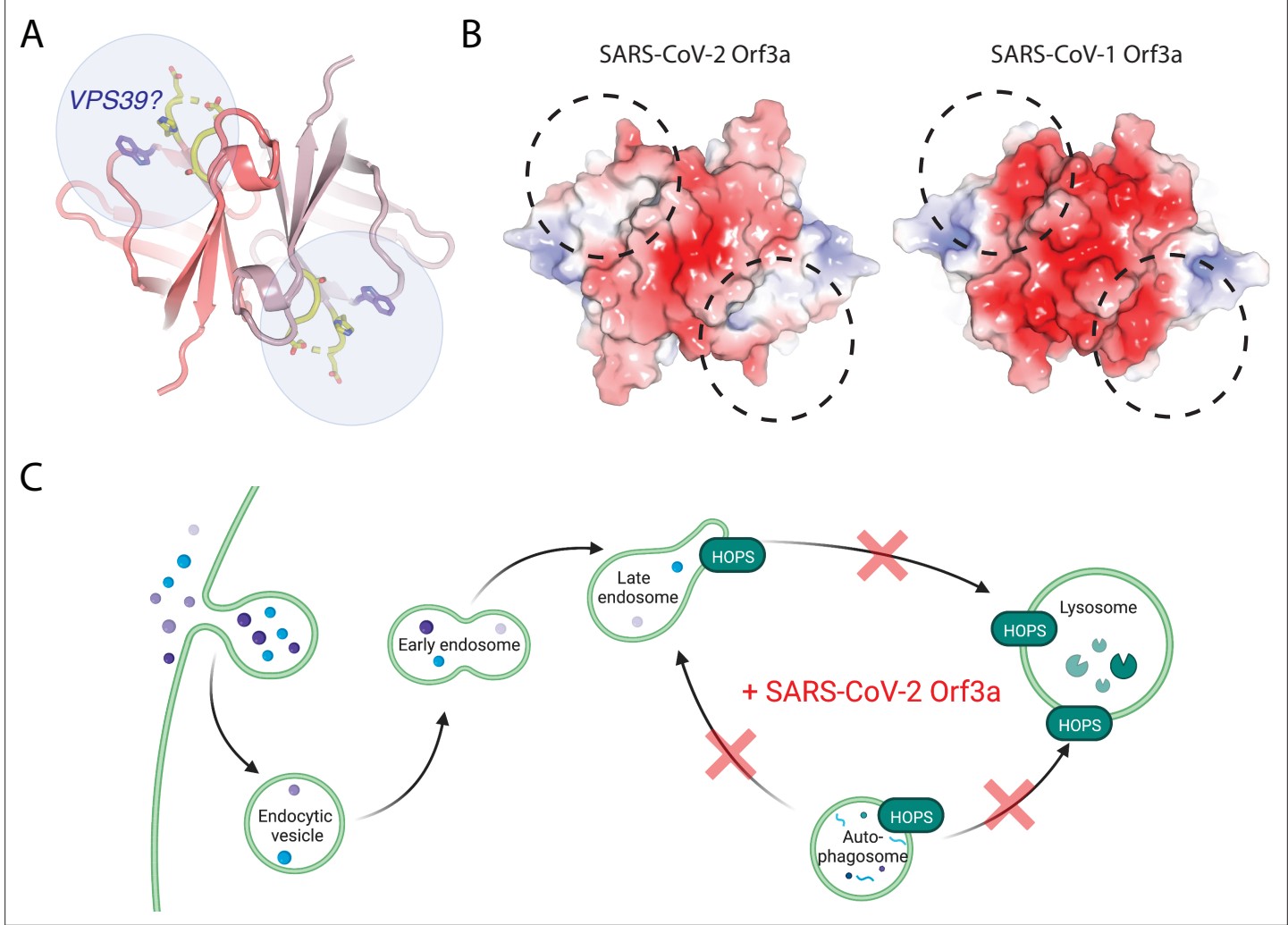

**Figure 7.** A region of VPS39 and SARS-CoV-2 Orf3a interaction. (**A**) Cytosolic view of SARS-CoV-2 Orf3a structure (dark and light pink) with the unstructured loop highlighted in yellow. W193 (purple, sticks) has also been described to mediate an interaction between SARS-CoV-2 Orf3a and VPS39 (**Chen et al., 2021**). Putative VPS39 interfaces are indicated (light blue spheres) with potential stoichiometries of 1:1 or 1:2 molecules of VPS39 to SARS-CoV-2 Orf3a. (**B**) Cytosolic surface of SARS-CoV-2 and SARS-CoV-1 Orf3a colored by their electrostatic potential (APBS program): blue, positive (+5 kT/e); red, negative (–5 kT/e) (**Jurrus et al., 2018**). The putative VPS39 interfaces (dotted black lines) are the same as indicated in (**A**). (**C**) Working model of SARS-CoV-2 Orf3a dysregulation of late endosome and autophagosome fusion with lysosomes. HOPS-dependent regions of the endocytic and autophagy pathways that are disrupted by SARS-CoV-2 Orf3a are indicated (red X). Adapted from "Mutation of HOPS Complex Subunits", by BioRender.com (2022). Retrieved from https://app.biorender.com/biorender-templates.

in HEK293 and A549 cells, and endolysosomal patch-clamp from HEK293 cells, in numerous cationic conditions. We also tested SARS-CoV-2 Orf3a at the PM of *Xenopus* oocytes and in reconstituted systems using purified SARS-CoV-2 Orf3a. We were unable to record currents that we can attribute to SARS-CoV-2 Orf3a.

While performing these studies, several other groups published electrophysiology data to either support or oppose the SARS-CoV-2 Orf3a viroporin hypothesis (**Kern et al., 2021**; **Toft-Bertelsen et al., 2021**; **Grant and Lester, 2021**). Our results from *Xenopus* oocytes are consistent with **Grant and Lester, 2021**. We propose that the reported SARS-CoV-2 Orf3a current observed by Toft-Bertelsen et al. could be due to a technical artifact such as pipette leak or contributed by endogenous channels (**Toft-Bertelsen et al., 2021**; **Harrison et al., 2022**; **Ackerman et al., 1994**). Similar to Kern et al., we were able to measure multiple large single channel-like conductances from vesicle-reconstituted SARS-CoV-2 Orf3a in $K^+$, $Na^+$, and $Ca^{2+}$ conditions using an atypically high protein to lipid ratio (1:10, wt:wt) (**Kern et al., 2021**; **Miller, 1986**). These channel-like conductances were not

present in samples reconstituted using a standard protein to lipid mixture of 1:100 (wt:wt) (*Miller, 1986*). The lack of macroscopic currents recorded from vesicles containing high protein amounts was concerning since we estimate that ~1,000 putative channels/$\mu m^2$ should be present (*Goldberg and Miller, 1991*; *Maduke et al., 1999*). We suspect that these channel-like conductances reflect spontaneous leak due to the high protein amounts in the sample, or possibly a contaminating channel, such as VDAC1 (*Niu et al., 2021*). Using several flux assays, we did not observe $K^+$ or $Cl^-$ flux from any of our reconstituted conditions, further indicating that the channel-like conductances are likely spurious.

To attempt to capture SARS-CoV-2 Orf3a in different conformational states, we determined three cryo-EM structures of SARS-CoV-2 Orf3a in lipid environments mimicking the PM and LE/Lyso and in a nanodisc containing a different membrane scaffold protein. We observe one nearly identical state in all SARS-CoV-2 Orf3a sample preparations, consistent with the published cryo-EM structure of SARS-CoV-2 Orf3a, determined at 2.1 Å resolution (global RMSD 0.38 Å between the SARS-CoV-2 Orf3a PM MSP1D1 nanodisc and published SARS-CoV-2 Orf3a structures, *Figure 3—figure supplement 3K*; *Kern et al., 2021*). We do not observe a membrane-spanning pore wide enough to permit the movement of dehydrated cations. Furthermore, the positive electrostatic potential within the aqueous vestibule would not favor cation permeation.

We also attempted to reproduce published data by investigating SARS-CoV-1 Orf3a viroporin activity in both HEK293 cells and *Xenopus* oocytes (*Lu et al., 2006*; *Chan et al., 2009*). We were unable to record cationic currents attributable to SARS-CoV-1 Orf3a in either system, yet we observed its localization at the PM. Similarly, we were unable to observe $K^+$ flux from vesicle-reconstituted SARS-CoV-1 Orf3a. We also determined a cryo-EM structure of SARS-CoV-1 Orf3a in an LE/Lyso MSP1D1-containing nanodisc and found its overall architecture indistinguishable from the SARS-CoV-2 homolog. From our comprehensive evaluation of both SARS-CoV-1 and SARS-CoV-2 Orf3a in multiple cell lines and reconstituted systems, and the conservation between the cryo-EM structures, we conclude that SARS-CoV-1 and SARS-CoV-2 Orf3a proteins are not viroporins.

Although a wide, membrane-spanning pore is not evident from the SARS-CoV-1 and SARS-CoV-2 Orf3a structures, we do observe density in our cryo-EM maps that we attribute to lipids. By comparing our four cryo-EM maps – three SARS-CoV-2 Orf3a samples reconstituted into nanodiscs containing PM- or LE/lyso-like lipid compositions and either MSP1D1 or Saposin A protein scaffolds, and one SARS-CoV-1 Orf3a sample reconstituted into a MSP1D1-containing nanodisc in a LE/Lyso-like environment – we detect two independent lipid binding sites likely occupied by DOPE. The cytosolic orientation of the two Orf3a fenestrations further supports the assignment of PE lipids since they are enriched in the inner leaflet of the PM and likely LE/Lyso (*van Meer et al., 2008*). We reason that Orf3a is not engaged with both molecules simultaneously due to steric clashes with R122 and between the two DOPE headgroups, as well as differences in lipid density presence or absence between the cryo-EM datasets. This is consistent with the published cryo-EM structure of SARS-CoV-2 Orf3a, in which the authors observed density in Lipid Site 1 and high-resolution density for the MSP1D1 scaffold protein, as we observe in our SARS-CoV-1 Orf3a map (*Kern et al., 2021*).

The possibility of two discrete sites for lipid binding may hint at a general function of Orf3a proteins that involves interaction with lipids in the bilayer. Intracellular vesicle formation is a cellular hallmark of a SARS-CoV-1 infection, possibly contributing to the region of viral RNA replication that is proposed to provide protection from host immune responses (*Freundt et al., 2010*). These intracellular vesicles are absent in Vero cells infected with SARS-CoV-1 ΔOrf3a, suggesting that Orf3a may be necessary for the formation of the vesicles (*Freundt et al., 2010*). Since the membrane vesicle structures are also observed during a SARS-CoV-2 infection, it is plausible that Orf3a may be required for this process and this has yet to be explored.

In addition to its interaction with lipids, we asked if there are host proteins that may bind to SARS-CoV-1 or SARS-CoV-2 Orf3a to better understand its possible contributions to viral pathogenesis. We turned to published proteomics datasets and focused on VPS39, a member of the HOPS membrane tethering complex that is essential for LE or AP fusion with Lyso (*Gordon et al., 2020a*; *Balderhaar and Ungermann, 2013*; *Gordon et al., 2020b*). A cellular defect common to HOPS complex mutants is an accumulation of LE and AP. We observe an enrichment of Rab7 puncta in cells overexpressing SARS-CoV-2 Orf3a$_{HALO}$, but not SARS-CoV-1 Orf3a$_{HALO}$. Consistent with this, VPS39 selectively interacts with SARS-CoV-2 Orf3a, and not SARS-CoV-1 Orf3a. By generating Orf3a chimeras (LC) we showed that an unstructured loop of SARS-CoV-2 Orf3a mediates binding to VPS39. Although SARS-CoV-1

Orf3a LC displays enhanced interaction with VPS39, it is not fully restored, and overexpression of the chimera is not sufficient to generate the Rab7 enrichment phenotype. Overall, our data suggests that the VPS39 interaction with Orf3a was acquired in SARS-CoV-2 and, by sequence and structural comparison between SARS-CoV-1 and SARS-CoV-2 Orf3a, that we have likely identified SARS-CoV-2 Orf3a's novel region of interaction.

Concurrent with our cellular and biochemical studies connecting SARS-CoV-2 Orf3a overexpression with HOPS complex fusion defects, several other groups published comprehensive work that support our findings (*Miao et al., 2021*; *Chen et al., 2021*; *Qu et al., 2021*). In particular, Chen et al. surveyed the cytosolic domain of SARS-CoV-2 Orf3a to find a region of VPS39 interaction and identified a similar binding surface (*Chen et al., 2021*). Two SARS-CoV-2 Orf3a residues, S171 and W193, were identified to be important for mediating this interaction (*Chen et al., 2021*). Our Orf3a loop chimeras include the S171 substitution and their results support our conclusion that SARS-CoV-1 Orf3a LC can partially rescue the VPS39 interaction, but it is not sufficient to restore the SARS-CoV-2 Orf3a Rab7 phenotype (*Chen et al., 2021*). Chen et al. demonstrate that a W193 substitution into SARS-CoV-1 Orf3a can partly restore their SARS-CoV-2 Orf3a cellular phenotype, further delineating the SARS-CoV-2 Orf3a binding surface (*Chen et al., 2021*). A comparison of the surface electrostatic potential in this putative region of VPS39 binding highlights an acidic region of SARS-CoV-1 Orf3a which is neutral on the SARS-CoV-2 Orf3a surface (*Figure 7B*). We propose that the unstructured loop, W193 substitution, and the neutral charge in this region contribute to promoting the SARS-CoV-2 Orf3a and VPS39 interaction (*Figure 7A-B*).

How does the acquired interaction between SARS-CoV-2 Orf3a and VPS39 contribute to viral pathogenesis? Several recent studies have implicated the endocytic pathway as the primary mode of viral egress for several β-coronaviruses, including SARS-CoV-2, and that Orf3a may mediate this process (*Ghosh et al., 2020*; *Chen et al., 2021*). One possibility is that the SARS-CoV-2 Orf3a and VPS39 interaction promotes viral exit by perturbing forward endocytic trafficking through disrupting the HOPS complex-mediated fusion of LE with Lyso (*Figure 7C*). This could be due to (1) Orf3a sequestering VPS39 from the HOPS complex or, (2) Orf3a interacting with the HOPS complex through VPS39, an important distinction that has yet to be elucidated. A second, related possibility is that the SARS-CoV-2 Orf3a and VPS39 interaction promotes viral egress by interacting with known membrane tethering, fusion and trafficking complexes that facilitate lysosome movement to the cell periphery. In particular, VPS39 knockdown in SARS-CoV-2 Orf3a overexpressing cells reduces the localization of LAMP1 vesicles near the PM, and concomitantly diminishes recruitment of protein complexes involved in Lyso-PM trafficking (*Chen et al., 2021*). These data suggest that VPS39 is necessary for SARS-CoV-2 Orf3a-mediated Lyso-PM trafficking and implicates the recruitment of the HOPS complex by Orf3a in this process (*Chen et al., 2021*).

A third possibility is the SARS-CoV-2 Orf3a and VPS39 interaction prevents HOPS-mediated AP-Lyso fusion, which we and others have observed (Miller and Clapham, not shown, *Figure 7C*; *Miao et al., 2021*; *Qu et al., 2021*). Autophagy is an intracellular surveillance process that targets damaged cellular or foreign materials, such as viruses, for lysosomal degradation. Many viruses hijack host cell autophagy to prevent its degradation and promote survival. Overall, the acquired SARS-CoV-2 Orf3a and VPS39 interaction may function to assist with SARS-CoV-2 exit and host intracellular immune evasion. The molecular details of the host cell response to SARS-CoV-2 Orf3a need to be examined during viral infection to fully elucidate the contributions of Orf3a to SARS-CoV-2 cell physiology and pathogenesis.

## Materials and methods
### Antibodies and dyes
SnapTag and HaloTag Janelia Fluor dyes were obtained from the Janelia Research Campus (*Grimm et al., 2017*). Rabbit monoclonal antibodies against EEA1 and against LAMP1 were from Cell Signeling Technologies (Catalog #: 3288 and 9091; RRID: AB_2096811 and AB_2687579). Mouse monoclonal antibody against Rab7 (Rab7-117) was from Sigma Aldrich (Catalog #: R8779; RRID: AB_609910). Rabbit polyclonal antibody against TGN46 was from Novus Biologicals (Catalog #: NBP1-49643; RRID: AB_10011762). Rabbit monoclonal antibody against GFP was from Abcam (Catalog #: ab32146, RRID: AB_732717). Mouse monoclonal antibody against the Strep tag (StrepMAB-Classic) was from IBA

Lifesciences (Catalog #: 2-1507-001; RRID: AB_513133). Hoechst 33342 dye was from Thermo Fisher (Catalog #: H3570). Alexa Fluor 488 goat anti-mouse and goat anti-rabbit antibodies were from Thermo Fisher (Catalog # A32723 and A32731; RRID: AB_2633275 and AB_2633280). HRP-conjugated mouse and rabbit antibodies were from Jackson ImmunoResearch Laboratories Inc (Catalog # 115-035-174 and 111-035-152; RRID AB_2338512 and AB_2337936).

## Molecular biology

Codon-optimized SARS-CoV-1 Orf3a was synthesized (GenScript) and codon-optimized SARS-CoV-2 Orf3a was a generous gift from Nevan Krogan (*Gordon et al., 2020b*). Human VPS39 was received from the MGC collection (Horizon Discovery). Full-length SARS CoV-1 Orf3a (Uniprot P59632) and SARS-CoV-2 Orf3a (Uniprot P0DTC3) were cloned into pcDNA3.1 for transient mammalian expression, a modified Piggyback vector for Tet-inducible expression, XLone, and a modified pEZT-BM vector for expression using the BacMam system (*Randolph et al., 2017*; *Morales-Perez et al., 2016*; *Goehring et al., 2014*). Both constructs were sub-cloned using *NheI* and *NotI* restriction sites into pcDNA3.1 and pEZT-BM vectors, or *MhuI* and *SpeI* restriction sites into the XLone vector. Constructs contained a C-terminal SNAP, HALO, or GFP moiety for cell imaging, or a Twin-Strep affinity purification tag that could be removed by HRV-3C protease. Full-length human VPS39 (Uniprot Q96JC1-2) was sub-cloned using *NheI* and *NotI* restriction sites into a modified pEZT-BM vector (*Morales-Perez et al., 2016*; *Goehring et al., 2014*). This construct contained a C-terminal GFP tag that could be removed by TEV protease. For cRNA generation, pcDNA3.1-containing SARS-CoV-1 Orf3a$_{2x-STREP}$ and SARS-CoV-2 Orf3a$_{2x-STREP}$ were linearized with the *XmaI* restriction enzyme. Linearized DNA was used as template for in vitro RNA synthesis (mMESSAGE mMACHINE T7 Transcription Kit; Thermo Fisher).

## Generation of HEK293T and A549 stable cell lines

HEK293T or A549 cell lines (ATCC, CRL-3216 and ATCC, CCL-185) were cultured in DMEM (ATCC) supplemented with 10% tetracycline negative fetal bovine serum (FBS, Gemini Bio Products) at 37 °C with 5% $CO_2$. For generation of stable lines, cells were detached using 0.25% Trypsin-EDTA (Thermo Fisher) and seeded at a density of $2x10^5$ and allowed to recover overnight. The following day, cells were transfected at a density of ~$2.5 \times 10^5$ using either Lipofectamine 2000 (HEK293T cells) or Lipofectamine 3000 (A549 cells; Thermo Fisher) with 2 µg of each Orf3a XLone constructs and 0.8 µg of the hyperactive piggyBac transposase. After 24 hr, cells that had successfully integrated Orf3a were selected with 10–30 µg/mL of blasticidin-HCl (Gemini Bio Products). Cells were passaged x2 before expanding and flash-cooling. Doxycycline-inducible expression was tested in each cell line using three different concentrations (0.1–1 µg/mL) of doxycycline hyclate (Sigma-Aldrich) and a no-doxycycline control to evaluate Orf3a expression leak. The cell line with the highest expression and minimal leak were selected for experiments (HEK293T cells: SARS-CoV-2 Orf3a$_{SNAP}$, SARS-CoV-2 Orf3a$_{HALO}$ LC and SARS-CoV-1 Orf3a$_{HALO}$ LC, A549 cells: SARS-CoV-2 Orf3a$_{GFP}$) or were sorted by flow cytometry (see below) to identify a population of cells that expressed Orf3a at a high concentration based on fluorescence intensity (HEK293T cells: SARS-CoV-2 Orf3a$_{HALO}$ and SARS-CoV1 Orf3a$_{HALO}$).

## Flow cytometry

Cells were trypsinized, pelleted, and then resuspended in 500 µl PBS and sorted into tubes or 96-well plates with a Sony SH800 cell sorter. The gating strategy to determine the SARS-CoV-2 Orf3a$_{HALO}$ and SARS-CoV1 Orf3a$_{HALO}$ expressing cell populations consisted of control cell samples not incubated with Janelia Fluor635 dye, control cells incubated with Janelia Fluor635 dye and doxycycline-treated cells incubated with Janelia Fluor635 dye. Samples were gated based on forward and backscatter and sorted for Janelia Fluor635 dye events in the top 3–4% of Cy5-A channel fluorescence (Cy5: 638 nm laser excitation, 665 nm with a 30 nm bandpass emission).

## Co-localization and immuno-staining experiments

35 mm (Cellvis) or eight-well (Thermo Fisher) glass bottom dishes were pre-coated with 0.1 mg/mL poly-D lysine (Thermo Fisher) for 1 hr at 37 °C. Cells were seeded at a density of 1–1.5x$10^5$ cells and allowed to recover overnight. For cellular marker overexpression experiments, cells were transfected at a density of ~$2 \times 10^5$ cells using Lipofectamine 2000 (Thermo Fisher) with 0.5–1 µg/µL of plasmid DNA containing Sec61-mEmerald, αMannosidase II-mEmerald, Rab5-mEmerald, Rab7-mEmerald,

Lamp1-mEmerald or Pex11-mNeonGreen. Orf3a expression was induced with 100 ng/mL doxycycline after 24 hr, and JF635 HALO ligand was added the following day for 1–2 hr at 37 °C. Cells were washed twice before being switched to Live Cell Imaging Solution (Thermo Fisher).

For immunostaining of cellular markers, Orf3a expression was induced with 100 ng/mL doxycycline. The following day, cells were incubated with JF635 HALO ligand 1–2 hr at 37 °C. Cells were then fixed with 4% paraformaldehyde (Electron Microscopy Sciences) for 30 min at room temperature, rinsed x3 with PBS + 0.01% sodium azide (PBS-A). Cells were permeabilized with PBS + 0.1% Triton-X100 for 30 min on a shaker, followed by a rinse with PBS and block with BlockAid blocking solution (Thermo Fisher) for 0.5–1 hr. Primary antibody was added using the following dilutions in BlockAid buffer: 1:200 EEA1 rabbit monoclonal, 1:500 Rab7 mouse monoclonal, 1:100 LAMP1 rabbit monoclonal, and 1:200 TGN46 rabbit monoclonal. Samples were incubated for 1 hr at room temperature. Cells were washed x3 with PBS-A. Secondary antibodies were added using the following dilutions in BlockAid buffer: 1:1000 Alexa Fluor 488 goat-anti-mouse or 1:1000 Alexa Fluor 488 goat anti-rabbit, and 1:5000 Hoescht 33342 stain. Samples were incubated for 1 hr at room temperature, washed and stored in PBS-A.

Images were acquired using a Zeiss 880 Laser Scanning Confocal microscope equipped with a Plan-Apochromat 63 x oil objective, 40 x multi-immersion LD LCI Plan-Apochromat objective and a 20 x air Plan-Apochromat objective. Data acquisition used ZEN Black software (Zeiss Instruments). Hoechst 33342 dye was excited by 405 nm laser light and the spectral detector set to 409–481 nm. mEmerald, mNeonGreen and GFP fluorophores were excited by 488 nm laser light and the spectral detector set to 490–545 nm. JF635 HALO ligand was excited with 633 nm light and the spectral detector set to 642–755 nm. The spectral detector was only used for non-Airyscan confocal scanning imaging sessions. Data analysis was performed in Image J. For cell images of Rab7 puncta, a median filter was applied to the confocal slices to remove background Rab7 staining prior to puncta quantification. Data processing of Airyscan confocal scanning images used auto settings for a 3D Z-Stack in ZEN Black (Zeiss Instruments). Statistical analyses of Rab7 puncta quantification were performed using estimation statistics (*Ho et al., 2019*).

## Total internal reflection microscopy

Cells were imaged using an inverted Nikon Eclipse TiS microscope main body and through-the-objective TIRF mode. A 488 nm solid-state laser (Coherent, Santa Clara, CA) was used for excitation. Excitation was controlled by a mechanical shutter (Uniblitz, VA Inc, Rochester, NY). The laser beam was focused to the backplane of a high- numerical aperture objective (60 x, N.A. 1.49, oil) by a combination of focusing lenses. Fluorescence emission was collected by an Orca Flash 4.0 sCMOS camera (Hamamatsu Photonics, Japan), after passing through an emission filter for the acquisition wavelength band of interest (540/40 nm, Semrock, US).

## Whole-cell electrophysiology in HEK293T cells

Doxycycline-inducible HEK293T cells, expressing $Orf3a_{SNAP}$ from SARS-CoV-1 or SARS-CoV-2, were cultured in DMEM supplied with 10% FBS (Gibco). 48 hr before electrophysiological recordings, cells were incubated in media with 100 ng/mL doxycycline, 500 nM JF505 SNAP ligand or vehicle. 1–3 hr before recording, cells were rinsed x3 with PBS and then maintained in fresh recording media. Whole-cell currents were obtained from transiently transfected HEK293T cells. Gigaseals were formed using 1–3 MΩ borosilicate pipettes (Warner Instruments). Whole-cell voltage clamp was performed; the macroscopic currents were acquired at 10 kHz and filtered at 5 kHz using an A-M Systems 2400 patch clamp amplifier. Series resistance and cell capacitance were compensated via amplifier controls. Current-voltage relationships were recorded using voltage ramps or steps protocols, with sweeps every 2 s. Currents are presented in pA/pF. Recordings were digitized using a Digidata 1320 A (Molecular Devices, LLC, California). The analysis was performed using Clampfit 10.3 (Molecular Devices, LLC, California).

For voltage ramp protocols performed in *Figure 2A–B*, internal solution contained (in mM) 150 KCl, 1 $MgCl_2$, 5 HEPES, pH 7.4, and 5 EGTA. External solution used for monovalent cations were (in mM): 150 XCl (X = $Na^+$, $K^+$, $Cs^+$ or $NMDG^+$), 5 HEPES, pH 7.4, 1 $MgCl_2$, and 1 $CaCl_2$, or 100 $CaCl_2$, 5 HEPES and 1 $MgCl_2$ at pH 7.4. For voltage steps protocols presented in *Figure 2C* and *Figure 2— figure supplement 3G–I*, internal solutions were (in mM) 147 KCl, 3 KOH, 1 $MgCl_2$ and 10 HEPES.

External solutions used were (in mM): 140 XCl (X = $Na^+$, $K^+$ or $Cs^+$), 10 HEPES, pH 7.4, 2 $MgCl_2$, 2 $CaCl_2$ and 8 glucose. No difference was detected from background currents measured in cells expressing SNAP-tagged and untagged versions of SARS-CoV-2 Orf3a (not shown).

## Whole-cell electrophysiology in A549 cells

Doxycycline-inducible A549 cells, expressing Orf3a$_{GFP}$ from SARS-CoV-2, were cultured in F-12 media (ATCC) supplied with 10% FBS (Gibco). After 24 h induction, whole-cell current analysis was performed using 3–5 MΩ pipettes containing (in mM): 10 NaCl, 130 KCl, 0.5 $MgCl_2$, 1 $CaCl_2$, 1 EGTA, and 5 HEPES, pH 7.4. The external solution used was comprised of (in mM): 136 NaCl, 4 KCl, 0.5 $MgCl_2$, 1 $CaCl_2$, 1 EGTA, and 5 HEPES, pH 7.4. After achieving whole-cell configuration, a ramp protocol was performed, as described in *Figure 2—figure supplement 2B–C* from –100 mV to +100 mV adjusted for liquid junction potential. Acquisition was performed using an Axopatch 200-B and was sampled at 10 kHz with a 5 kHz lowpass Bessel filter. Pipette capacitance was compensated to 80%. Current was normalized via cell capacitance and reported as current density.

## Endolysosomal patch-clamp

HEK293T cells were cultured in DMEM (Thermo Fisher) supplemented with 10% FBS (R&D Systems) and 1×penicillin–streptomycin (Thermo Fisher) at 37 °C with 5% $CO_2$. HEK293T cells were plated on 12 mm poly-l-lysine-coated coverslips in 24 well plates, and ORF3a-HALO was transfected using PolyJet (SignaGen). The medium was replaced with fresh DMEM containing 10% FBS and 1×penicillin–streptomycin after 6 hr of transfection, followed by 1 µM vacuolin to enlarge endolysosomes. Whole-endolysosomal recordings were performed as previously described with an Axopatch 200-B amplifier and Digidata 1440 A controlled by pClamp and Clampfit (Molecular Devices) (*Cang et al., 2015*; *Chen et al., 2017*). The recordings were done 24–32 hr after transfection. For recordings presented in *Figure 2D–I*, the bath solution contained (in mM): 75 KCl, 75 NaCl, 1 $MgCl_2$, 0.25 $CaCl_2$, 1 EGTA, 10 HEPES (pH 7.2). The pipette solution contained (in mM): 75 NaOH, 70 KOH, 5 KCl, 150 MSA (methanesulfonic acid), 1 $MgCl_2$, 1 $CaCl_2$, 10 MES (2-(N-morpholino)ethanesulfonic acid, pH 4.6) (*Figure 2D–F*) or 50 $Ca(OH)_2$, 70 NaOH, 5 NaCl, 150 MSA, 1 $MgCl_2$, 1 $CaCl_2$, 10 MES (pH 4.6) (*Figure 2G–I*). For recordings presented in *Figure 2—figure supplement 1A-C*, the bath and pipette solution contained (in mM) 150 NMDG, 1 HCl, 10 HEPES, 10 MES (pH 4 or pH 7 by MSA).

## Two-electrode voltage-clamp

Defolliculated *Xenopus laevis* oocytes were purchased from Ecocyte Bioscience, injected with either water or ~20 ng cRNA per oocyte, and stored at 17 °C in ND96 buffer (in mM): 96 NaCl, 2 KCl, 5 HEPES, 1 $MgCl_2$ and 1.8 $CaCl_2$, pH 7.5 with NaOH, supplemented with 50 µg/ml gentamycin and 1% Foundation FBS (Gemini Bio Products). Recordings were performed 2–3 days after injection in ND96 or high $K^+$ buffer (in mM): 2 NaCl, 96 KCl, 5 HEPES, 1 $MgCl_2$ and 1.8 $CaCl_2$, pH 7.5 with KOH at room temperature using a Warner OC-725C Oocyte Clamp amplifier (Warner Instrument Corp, USA) in a ~500 µL recording chamber. Recording microelectrodes were pulled from borosilicate glass using a P-97 Puller System (Sutter Instrument) with resistances of 0.2–0.5 MΩ, then filled with 3 M KCl. Data were acquired using pCLAMP 10 software (Molecular Devices) and a Digidata 1440 A digitizer (Molecular Devices), filtered at 0.5 kHz and digitized at 10 kHz. Oocytes were subject to 200ms steps from –135 mV to +60 mV in +15 mV increments applied every 5 s, from a holding voltage of 0 mV. The absolute peak current from within the step was analyzed.

## Surface biotinylation

Three days after oocyte injections, 20–25 oocytes per sample were washed x6 in ND96 and labeled with 0.5 mg/ml EZ-Link Sulfo-NHS-SS-Biotin (Thermo Fisher) for 30 min at room temperature. 'No biotinylation' samples were incubated without the biotinylation reagent. Oocytes were washed again x6 in ND96 before homogenizing (by pipetting up and down 15 times with a P200 pipette) in 500 µL lysis buffer (1% Triton X-100, 100 mM NaCl, 20 mM Tris-HCl, pH 7.4) plus 1:1000 Protease Inhibitor Cocktail Set III, EDTA-Free (Calbiochem). All subsequent steps were performed at 4 °C. Lysates were gently shaken for 10 min then centrifuged at 16,000 x *g* for 5 min. A 50 µL aliquot of the supernatant (total cell protein) was stored at –80 °C for later use. The remaining supernatant was diluted 1:1 with the lysis buffer, then 100 µL of NeutrAvidin agarose beads (Thermo Fisher) added and the sample

shaken gently overnight at 4 °C. Beads were washed x6 in 500 μL lysis buffer with a 2 min 6000 x $g$ centrifugation between each wash, and finally resuspended in 50 μL lysis buffer. Both this sample and the total cell protein sample were mixed 1:1 with 2 x loading buffer: 50% 4 x Laemmli Sample Buffer (Bio-Rad), 30% 100 mM DTT, 10% 2-mercaptoethanol, and 10% Buffer BXT (IBA Lifesciences) and gently rotated for 1 hr at room temperature. Samples were centrifuged for 2 min at 12,000 x $g$ before being separated on 8–16% Mini-PROTEAN TGX Precast Protein Gels (Bio-Rad) at 160 V for 60 min. PageRuler Plus Prestained Protein Ladder (Thermo Fisher) was used as the size standard. Samples were transferred onto a polyvinylidene difluoride membrane activated with methanol using the Trans-Blot Turbo Transfer System (Bio-Rad). Membranes were probed with mouse StrepMAB-Classic antibody diluted 1:3000 in TBS-T (25 mM Tris, 137 mM NaCl, 3 mM KCl, 0.05% Tween 20), followed by horseradish peroxidase conjugated anti-mouse secondary antibody diluted 1:10,000 in TBS-T, then developed using SuperSignal West Pico PLUS Chemiluminescent Substrate (Thermo Fisher).

## Baculovirus generation and expression of SARS-CoV-1 Orf3a$_{2x-STREP}$ and SARS-CoV-2 Orf3a$_{2x-STREP}$

To produce high-titer SARS-CoV-1 Orf3a$_{2x-STREP}$, SARS-CoV-2 Orf3a$_{2x-STREP}$ and VPS39$_{GFP}$ baculovirus, plasmids were first transformed into DH10Bac *E. coli* (Thermo Fisher) to generate bacmid DNA, using a blue-white colony selection strategy. Bacmid DNA was purified and transformed using CellFectin II reagent (Thermo Fisher) into Sf9 cells resuspended in fresh ESF 921 Insect Cell Culture Medium, Protein Free (Expression Systems) and seeded in a six-well plate to produce baculovirus. After 3- to 4-day incubation at 28 °C (<70% cell viability and abundant GFP expression), the supernatant containing a low titer of baculovirus (P0 virus) was collected and filtered. P0 virus was used to infect a larger suspension (100 mL) of Sf9 cells to increase baculovirus titer (P1 virus), collected and filtered after 4 days of incubation at 28 °C. A final round of baculovirus amplification in Sf9 cells yielded 200–300 mL of baculovirus (P2 virus), supplemented with 5% Foundation FBS (Gemini Bio Products).

For protein expression, SARS-CoV-1 Orf3a or SARS-CoV-2 Orf3a P2 virus (3–10% of total cell suspension volume) was used to infect HEK293S GnTI- cells, grown in FreeStyle 293 Expression Medium (Thermo Fisher) adapted to suspension growth in 2% Foundation FBS (Gemini Bio Products). Cells were incubated in suspension for 24 hr at 37 °C. Sodium butyrate (10 mM, Sigma-Aldrich) was added to enhance protein expression, and the cells were left for another 2–3 days incubating in suspension at 37 °C. Cells were harvested by pelleting at 400 x $g$, flash-cooled and stored at –80 °C. For cells overexpressing VPS39$_{GFP}$, a 100 mL suspension was split into 10 mL aliquots and harvested as described.

## Purification of SARS CoV-1 and SARS-CoV-2 Orf3a

Cells (0.5–1 L biomass) were agitated for 1 hr at 4 °C in Purification Buffer containing 100 mM Tris-HCl, pH 8.0, 75 mM NaCl, 75 mM KCl, 1 mM EDTA, pH 8.0, and supplemented with 25 μg/mL deoxyribonuclease I (DNase, Gold Biotechnology), a 1:1000 dilution of Protease Inhibitor Cocktail Set III, EDTA-free (PI Mix III, Calbiochem), 1 mM 4-(2-Aminoethyl)-benzenesulfonylfluoride hydrochloride (AEBSF, Gold Biotechnology), 1 mM benzamidine hydrochloride monohydrate (benzamidine, Gold Biotechnology), and 30 mM n-dodecyl-β-D-maltopyranoside (DDM, Anatrace). Following membrane protein extraction, the soluble fraction was clarified by centrifugation at 40,000 x $g$ for 45 min at 4 °C and vacuum filtered. For affinity purification with the twin-strep tag (22 °C), lysate was sequentially applied over two columns packed with 1–2 mL of Strep-Tactin XT resin (IBA-Lifesciences), pre-equilibrated in Wash Buffer (Purification Buffer supplemented with 1 mM DDM). After application of lysate, the resin was rinsed with ~5 column volumes of Wash Buffer and protein was eluted after a 30 min incubation of resin in 3 mL of Buffer BXT (IBA-Lifesciences) supplemented with 75 mM KCl and 1 mM DDM. The collected elution was concentrated to 6 mg/mL using a 50,000 Da concentrator (Amicon Ultra, EMD Millipore) and further purified by SEC using a Superdex 200 Increase 10/300 GL (Superdex 200, GE Healthcare) column in SEC buffer containing 20 mM HEPES, pH 7.5, 75 mM NaCl, 75 mM KCl, and supplemented with 3 mM n-decyl-β-D-maltopyranoside (DM, Anatrace).

## Expression and purification of MSP1D1

The nanodisc scaffold protein MSP1D1 was expressed and purified as described previously with modifications (*Ritchie et al., 2009*). MSP1D1 pET28a was transformed and expressed in *E. coli*

B21-CodonPlus (DE3)-RIPL competent cells (Agilent). MSP1D1 contains a TEV protease-removable N-terminal 6xHis tag. For purification, MSP1D1 *E. coli* pellets were agitated for 20 min at room temperature in buffer (25 mL/1 L of biomass) containing 20 mM sodium phosphate, pH 7.4 supplemented with 25 µg/mL DNase, a 1:1000 dilution of PI Mix III, 1 mM AEBSF, and 1 mM benzamidine. The resuspension was lysed by passing it x4 through a microfluidizer EmulsiFlex-C5 (Avestin) and clarified by centrifugation at 40,000 x *g* for 40 min at 4 °C. For affinity purification, 300 mM NaCl was added to the lysate, combined with Ni-NTA resin (2 mL/1 L of biomass, Qiagen) pre-equilibrated with 40 mM sodium phosphate buffer, pH 7.4 and rotated for 1 hr at 4 °C. After batch binding and packing of the Ni-NTA resin, all subsequent wash and elution steps follow as described previously (*Ritchie et al., 2009*).

To remove the 6xHis tag fused to MSP1D1, TEV protease was added to the elution fraction at a 1:20 wt:wt ratio, supplemented with 0.5 mM EDTA. The next day, TEV protease cleavage was confirmed by SDS-PAGE and the sample was dialyzed overnight in buffer containing 40 mM Tris-HCl, pH 8.0 and 300 mM NaCl. The following day, residual 6xHis-tagged TEV protease was removed by the addition of Ni-NTA resin, and the sample was incubated for 2 hr at room temperature. After removing the resin, the sample was dialyzed overnight in storage buffer containing 20 mM HEPES, pH 7.4, 75 mM NaCl, 75 mM KCl and 0.2 mM DDM. The next day, the final sample was concentrated to ~5 mg/mL in a 10,000 Da concentrator (Amicon Ultra, EMD Millipore), flash-cooled and stored at –80 °C.

## Expression and purification of Saposin A

Saposin A was expressed and purified as described previously with modifications (*Frauenfeld et al., 2016*; *Flayhan et al., 2018*). Saposin A pET28a was transformed and expressed in *E. coli* Rosetta-gami 2 (DE3) competent cells (Thermo Fisher). Saposin A contains a TEV protease-removable N-terminal 6xHis tag. For purification, Saposin A *E. coli* pellets were agitated for 20 min at 4 °C in buffer (25 mL/1 L of biomass) containing 20 mM HEPES, pH 7.5, 150 mM NaCl, 20 mM Imidazole and 5% glycerol, supplemented with 25 µg/mL DNase, a 1:1000 dilution of PI Mix III, 1 mM AEBSF, and 1 mM benzamidine. The resuspension was lysed by passing it x4 through a microfluidizer and first clarified by centrifugation at 2000 x *g* for 30 min at 18 °C to remove insoluble cell debris. The sample was heated at 85 °C for 10 min, followed by a second clarification step by centrifugation at 55,000 x *g* for 45 min at 18 °C. For affinity purification, supernatant was combined with Ni-NTA resin (1 mL/1 L of biomass, Qiagen) pre-equilibrated with 20 mM HEPES, pH 7.5 and 150 mM NaCl and rotated for 1 hr at 4 °C. After batch binding and packing of the Ni-NTA resin, all subsequent wash and elution steps follow as described previously (*Frauenfeld et al., 2016*).

To remove the 6xHis tag fused to Saposin A, TEV protease was added to the elution fraction at a 1:10 wt:wt ratio. The sample was dialyzed overnight in a 3.5 kDa molecular weight cutoff dialysis cassette (Slide-a-Lyzer, Thermo Fisher) at 4 °C against 20 mM HEPES, pH 7.5 and 150 mM NaCl. The next day, the sample was passed over a Ni-NTA column to remove uncleaved Saposin A and 6xHis-tagged TEV protease, followed by SDS-PAGE to confirm cleavage. This sample was dialyzed in the same type of dialysis cassette overnight in storage buffer containing 20 mM HEPES, pH 7.5, 75 mM NaCl and 75 mM KCl. The next day, 0.2 mM DDM was added prior to concentration to prevent aggregation. The final sample was concentrated to ~2.8 mg/mL (10,000 Da concentrator, Amicon Ultra, EMD Millipore), flash-cooled and stored at –80 °C.

## Orf3a reconstitution into MSP1D1 and Saposin A nanodiscs

SARS-CoV-1 Orf3a and SARS-CoV-2 Orf3a were reconstituted into MSP1D1 nanodiscs containing two distinct lipid compositions which we refer to as 'plasma membrane' (PM) or 'late endosomal/lysosomal' (LE/Lyso). A 2:1:1 (wt:wt) PM lipid mixture of 1,2-dipalmitoleoyl-sn-glycero-3-phosphocholine, 1,2-dioleoyl-sn-glycero-3-phosphoethanolamine, and 1,2-dioleoyl-sn-glycero-3-phospho-L-serine (16:1 PC:DOPE:DOPS, Avanti Polar Lipids) or a 4:2:1 (wt:wt) LE/lysosome lipid mixture of 16:1 PC, DOPE, and (S,S) bisoleoyl-lysobisphosphatidic acid (LBPA, Echelon Biosciences) was prepared at 10 mM in SEC buffer with 5 mM DM. For MSP1D1 nanodiscs, lipid nanodiscs were formed by first combining Orf3a, MSP1D1 and the lipid mixture together at a molar ratio of 1:4:260 and rotating at 4 °C for 1 hr. For Saposin A nanodiscs, a molar ratio of 1:15:40 of Orf3a:Saposin A:lipid was used to form protein-reconstituted nanodiscs. First, SARS-CoV-2 Orf3a was added to the lipid mixture and

incubated at room temperature for 10 min. Saposin A was then added to the mixture and incubated for 30 min at room temperature. For both MSP1D1 and Saposin A nanodiscs, detergent was removed to form lipid nanodiscs by the addition of Bio-beads (Bio-beads SM2 resin, Bio-rad) pre-equilibrated in SEC buffer (~1:3, wt:vol). Bio-beads were added and incubated with the sample rotating at 4 °C for 1 hr. Then, a second round of Bio-bead binding was performed, and the sample was left rotating at 4 °C overnight. The next day, the lipid nanodiscs were purified by SEC in detergent-free SEC buffer. The peak fraction in each case was collected, concentrated to 1.3–1.5 mg/mL using a 100,000 Da concentrator (Amicon Ultra, EMD Millipore), and immediately used for cryo-EM grid preparation.

## Cryo-EM sample preparation and data acquisition

Purified Orf3a sample (3 µL) was pipetted onto Quantifoil R1.2/R1.3 holey carbon grids (Cu 400, Electron Microscopy Sciences) which had been glow discharged for 1 min with a current of 15 mA using a PELCO easiGlow glow discharge cleaning system (Ted Pella). A vitrobot Mark IV cryo-EM sample plunger (FEI) (operated at 4 °C with a 4–5 s blotting time at a blot force of 3–4 and 100% humidity) was used to plunge-freeze the sample into liquid nitrogen-cooled liquid ethane. Grids were first imaged on a 200 keV Technai F20 TEM (FEI) equipped with a K2 Summit direct electron detector (Gatan, Inc) to evaluate sample quality, particle quantity and distribution. They were subsequently clipped and loaded into a 300 keV Titan Krios microscope (FEI) equipped with a Cs corrector to reduce spherical aberration of the objective lens from 2.7 mm to ~0.01 mm, BioQuantum energy filter (Gatan, Inc) and a K3 camera (Gatan, Inc). Images were recorded with SerialEM in super-resolution mode at a calibrated magnification of 59,242 x, which corresponds to a super-resolution pixel size of 0.422 Å, a defocus range of –0.8 to –2 µm, and a total e$^-$ dose of 50 e$^-$ per Å$^2$ (*Mastronarde, 2005*). Beam-image shift was used to acquire data from multiple regions of interest for one stage position to improve the throughput of data acquisition. Movies were saved without applying camera gain and with LZW compression to reduce file size.

## Cryo-EM data processing

*Figure 3—figure supplements 1 and 2*, *Figure 3—figure supplement 4* and *Figure 4—figure supplements 2 and 4* describe the workflow for SARS-CoV-2 Orf3a LE/Lyso MSP1D1 nanodisc, SARS-CoV-2 Orf3a PM MSP1D1 nanodisc, SARS-CoV-2 Orf3a LE/Lyso Saposin A nanodisc and SARS-CoV-1 Orf3a LE/Lyso MSP1D1 nanodisc datasets. Movie stacks were gain-corrected, two-fold Fourier-cropped to a calibrated pixel size of 0.844 Å/pix and dose weighted using the CPU-based Relion 3.0 (SARS-CoV-2 Orf3a LE/Lyso MSP1D1 nanodisc) or Relion 3.1 (SARS-CoV-2 Orf3a PM MSP1D1 nanodisc, SARS-CoV-2 Orf3a LE/Lyso Saposin A nanodisc and SARS-CoV-1 Orf3a LE/Lyso MSP1D1 nanodisc) implementations of MotionCor2 (*Zivanov et al., 2018*; *Zheng et al., 2017*). Contrast Transfer Function (CTF) estimates for motion-corrected micrographs were performed in CTFFIND4 using all 50 frames (*Rohou and Grigorieff, 2015*).

## SARS-CoV-2 Orf3a LE/Lyso MSP1D1 nanodisc dataset

Two datasets of the SARS-CoV-2 Orf3a LE/Lyso nanodisc preparation (SARS-CoV-2 Orf3a LE/Lyso datasets A and B) were collected with a total of 8442 and 5645 micrographs, respectively. Micrographs with CTF estimation fits worse than 5 Å or with a prominent ice ring as manually inspected from the CTF 2D power spectra were discarded, leaving a total of 6759 and 3033 good quality micrographs per dataset.

   Initial data processing was performed with SARS-CoV-2 Orf3a LE/Lyso dataset A. To assess the oligomeric state of SARS-CoV-2 Orf3a, all steps during the initial cycle of data processing were performed without symmetry applied. A total of 1123 particles were manually selected for reference-free 2D classification to generate templates that were used for automatic particle picking with a subset (~300) of micrographs in Relion 3.0 (*Zivanov et al., 2018*). Approximately 100,000 particles were selected and were used to improve the templates generated by reference-free 2D classification. The improved templates were used for automatic particle picking with the entire dataset. A total of 2,520,325 particles were subjected to two rounds of reference-free 2D classification in cryoSPARC v3.0 to remove outlier particles (e.g. remove ice contaminants), yielding 1,042,957 particles (*Punjani et al., 2017*). An initial model of SARS-CoV-2 Orf3a was generated by ab initio reconstruction in cryoSPARC v3.0 (*Punjani et al., 2017*). Three rounds were performed using low- and high-resolution

cutoffs of 9 Å and 7 Å, initial and final particle batch sizes of 1000 and 2000 particles, and a class similarity of 0.01 for each iteration. Symmetry was not applied at this stage. Using non-uniform refinement in cryoSPARC v3.0, the best initial 3D model from ~86,126 particles refined to ~6 Å and showed clear twofold symmetry. (*Punjani et al., 2020*).

To further improve the model, additional particles from the second round of reference-free 2D classification were selected (1,363,667 particles in total) and subjected to another round of 2D classification in cryoSPARC v3.0. A similar strategy of four rounds of ab initio reconstruction was performed, with the final iteration using an initial and final particle batch size of 2000 and 4000 particles, and a class similarity of 0.4. The best ab initio model from the last iteration was refined to 5.3 Å from 73,377 particles using non-uniform refinement with C2 symmetry applied. The particles were then subjected to Bayesian polishing in Relion 3.0 and refined in cryoSPARC v3.0, to yield a final reconstruction of similar resolution (*Zivanov et al., 2018*).

A data processing 'decoy' refinement strategy was implemented at this step to improve map resolution as described previously (*Liu et al., 2022*). Micrographs were over-picked with particles using a Laplacian of Gaussian filter in Relion 3.0 (*Zivanov et al., 2018*). Particles were subjected to a round of reference-free 2D classification in cryoSPARC v3.0 to remove ice contaminants. The starting particle stack of 6,145,296 was used to generate seven junk 'decoy' models from a cycle of ab initio reconstruction. The decoy maps and best SARS-CoV-2 Orf3a map (~5.3 Å) were used to sort false positive from Orf3a particles using heterogeneous refinement in cryoSPARC v3.0 (Round 1) (*Punjani et al., 2017*). Orf3a particles were subjected to additional rounds of heterogeneous refinement until most particles classified with the SARS-CoV-2 Orf3a model, with the final particle stack containing 1,565,551. Iterative rounds of ab initio reconstruction performed to further classify this subset of particles using similar parameters to what was described above. The best 3D ab initio model from 300,114 particles was refined to 4.1 Å using non-uniform refinement in cryoSPARC v3.0. Particles that had an inter-particle distance of <100 Å (particle duplicates) were removed and subjected to Bayesian polishing and 3D refinement in Relion 3.0. The final 3D reconstruction resolved to 3.8 Å.

Another iteration (Round 2) of the 'decoy' refinement strategy was performed with the starting particle stack. Three rounds of heterogeneous refinement using the improved SARS-CoV-2 Orf3a model at 3.7 Å were completed, followed by an additional reference-free 2D classification in cryoSPARC v3.0 with the 1,184,876 particles to remove outliers (*Punjani et al., 2017*). The selected particles from 2D classification (1,049,046 particles – eventually combined with dataset B) yielded a refined 3D reconstruction of ~4 Å. Particle duplicates were removed and the remaining particles were subjected to Bayesian polishing and a round of 3D refinement in Relion 3.0. To assess distinct conformational states, we performed 3D classification in Relion 3.0 using the consensus reconstruction as an initial model (low-pass filtered at 5 Å resolution) and applied an Orf3a-only mask. The particles were sorted into three to four classes using the fixed angular assignments from 3D refinement. Although this strategy did not yield a different conformation of Orf3a, it did help to identify a set of 186,327 particles that generated a reconstruction which refined to 3.8 Å with more 3D features.

At this stage, SARS-CoV-2 Orf3a LE/Lyso dataset B was combined with dataset A. Particles were automatically selected in dataset B using the Laplacian of Gaussian filter strategy in Relion 3.0. The dataset B particles were then subjected to the 'decoy' refinement strategy using the improved model of Orf3a (3.8 Å). The particles that were classified with the SARS-CoV-2 Orf3a model (962,666 in total) were combined with the 1,049,046 particles from dataset A, refined to ~4 Å resolution. These were processed with Bayesian polishing in Relion 3.0 and imported to cryoSPARC v3.0 for sorting by 2D classification and ab initio reconstruction. The best model reconstructed from 520,044 particles using non-uniform 3D refinement reached a resolution of 3.5 Å. The parameters from Bayesian polishing were recalculated for this subset of particles, and the particles were subjected a round of 3D refinement and 3D classification with the regularization parameter T adjusted to a value of 20 to sort out the best model at 3.4 Å resolution reconstructed from 119,318 particles.

A final iteration of data processing was performed using Topaz, a neural network particle picking program implemented in cryoSPARC v3.0 (*Bepler et al., 2019*). Topaz denoise was first used to reduce noise in all micrographs prior to particle picking (*Bepler et al., 2020*). The best 119,318 particles from the last round of data processing were used to train the ResNet8 neural network to select particles from SARS-CoV-2 Orf3a LE/Lyso datasets A and B (*Bepler et al., 2019*). The 7,557,588 particles selected were subjected to reference-free 2D classification to first remove ice contamination, followed

by a 'decoy' refinement strategy using the best 3.4 Å resolution model. The particles that sorted with the SARS-CoV-2 Orf3a model (1,430,051 in total) were further sorted by one round of ab initio reconstruction to identify the best subset of particles. The final reconstruction from a total of 679,097 particles in the best ab initio class refined to 3.4 Å resolution. Particle duplicates were removed, and the remaining particles were subjected to Bayesian polishing and a round of 3D refinement in Relion 3.0. This was followed by 3D classification in searching for 3 classes, with the regularization parameter T adjusted to a value of 40. The best map and particles from 3D classification were used for 3D refinement, CTF refinement, and a final round of 3D refinement in Relion 3.0 to yield a final map of 3.0 Å resolution.

## SARS-CoV-2 Orf3a PM MSP1D1 nanodisc dataset

A dataset of the SARS-CoV-2 Orf3a PM nanodisc preparation was collected with a total of 17,828 micrographs. Micrographs with CTF estimation fits less than 5 Å, a rlnCtfAstimatism value greater than 1000 Å, rlnFigureofMerit values below 0.1, or with a prominent ice ring as manually inspected from the CTF 2D power spectra, were discarded, leaving a total of 15,637 good quality micrographs (*Goehring et al., 2014*). Particle picking was performed using Topaz implemented in cryoSPARC v3.0 (*Bepler et al., 2019*). The micrographs were first denoised and 1678 particles were manually selected to train the ResNet8 neural network for particle selection (*Bepler et al., 2019*; *Bepler et al., 2020*). The 8,607,878 particles selected were subjected to reference-free 2D classification to first remove ice contamination, and were then sorted through iterative rounds of heterogeneous refinement with 7 'decoy' maps and the a 3.6 Å resolution map of the SARS-CoV-2 Orf3a LE/Lyso MSP1D1 nanodisc (*Chen et al., 2017*). An initial model from the particles which sorted into the 'good class' (475,773 particles) was generated by ab initio reconstruction and refined to 3.8 Å resolution using non-uniform refinement in cryoSPARC v3.0 (*Chen et al., 2017*). The initial model was mildly improved in Relion 3.1, through Bayesian polishing, 3D refinement and particle sorting by 3D classification, searching for three classes with fixed angular distribution and T=40 parameters (*Goehring et al., 2014*). A final reconstruction of 3.7 Å from 129,484 particles was used as input for the next round of particle picking and heterogeneous refinement.

The 129,484 particles from the first iteration Topaz particle picking and map improvement were used to train the ResNet8 neural network in Topaz. A total of 13,224,923 particles were selected and sorted by 2D classification and iterative rounds of 3D heterogeneous refinement with the decoy and SARS-CoV-2 Orf3a PM MSP1D1 nanodisc 3.7 Å maps. A total of 595,295 particles were selected and used for non-uniform refinement in cryoSPARC v3.0, which generated a 3.7 Å reconstruction (*Punjani et al., 2020*). This was improved in Relion 3.1 with Bayesian polishing, 3D refinement, two rounds of 3D classification in searching for four classes each time by fixing the angular distribution and T=40 parameters. The final model, reconstructed from 125,678 particles, was subjected to 3D refinement, CTF refinement, another round of Bayesian polishing, and 3D refinement to yield a final reconstruction of 3.4 Å resolution.

## SARS-CoV-2 Orf3a LE/Lyso Saposin A nanodisc dataset

Two datasets of the SARS-CoV-2 Orf3a Saposin A nanodisc preparation was collected with a total of 17,900 micrographs. Micrographs with CTF estimation fits worse than 5 Å, a rlnCtfAstimatism value greater than 1000 Å, a rlnFigureofMerit value that was below 0.1, or with a prominent ice ring as manually inspected from the CTF 2D power spectra were discarded, leaving a total of 15,946 good quality micrographs (*Goehring et al., 2014*). Particle picking was performed using Topaz (*Bepler et al., 2019*). The micrographs were first denoised, and 1297 particles were manually selected to train the ResNet8 neural network to select particles (*Bepler et al., 2019*; *Bepler et al., 2020*). The 12,070,379 particles selected were subjected to reference-free 2D classification to first remove ice contamination (*Chen et al., 2017*). An initial model of a SARS-CoV-2 Orf3a LE/Lyso Saposin A nanodisc was generated, first by ab initio reconstruction using cryoSPARC v3.0, using the same parameters described above and 8,303,729 particles (*Chen et al., 2017*). This was followed by non-uniform refinement in cryoSPARC v3.0 with the best 3D map from ab initio reconstruction. 1,907,669 particles refined to ~3.1 Å showed clear low-resolution density for 6 Saposin A molecules (*Punjani et al., 2020*).

This model was used with 7 decoy models to further parse out the best particles from a 11,247,076 stack using iterative heterogeneous refinement (*Chen et al., 2017*). The best particles were subjected

to 2D classification to further remove ice, another round of ab initio reconstruction, and the best map and particles from the best three classes generated a reconstruction to ~2.9 Å resolution containing 2,559,504 particles. To improve the map and look for other conformational states of SARS-CoV-2 Orf3a, we performed subsequent data processing steps in Relion 3.1 (*Goehring et al., 2014*). Duplicate particles were removed and then subjected to Bayesian polishing and 3D refinement. Next, two individual rounds of 3D classification were performed, searching for 6 or 10 classes with fixed angular distribution and T=40 parameters and an initial low resolution pass filter of 8 Å or 5 Å, respectively. We were unable to identify any alternative conformations. The best map resolved to 2.8 Å resolution after CTF refinement, 3D refinement, another round of Bayesian polishing and a final round of 3D refinement.

## SARS-CoV-1 Orf3a LE/Lyso nanodisc datasets

A dataset of the SARS-CoV-1 Orf3a LE/Lyso nanodisc preparation was collected with a total of 15,584 micrographs. Micrographs with CTF estimation fits >5 Å, a rlnCtfAstimatism value greater than 1000 Å, rlnFigureofMerit value that was below 0.1, or with a prominent ice ring as manually inspected from the CTF 2D power spectra were discarded, leaving 13,970 good quality micrographs. Particle picking was performed using Topaz (*Bepler et al., 2019*). The micrographs were first denoised, and 1504 particles were manually selected to train the ResNet8 neural network to select particles (*Bepler et al., 2019*; *Bepler et al., 2020*). The 4,697,964 particles selected were subjected to reference-free 2D classification to first remove ice contamination (*Chen et al., 2017*). An initial model of a SARS-CoV-1 Orf3a LE/Lyso MSP1D1 nanodisc was generated, first by two rounds of ab initio reconstruction using cryoSPARC v3.0 with 2,702,734 particles and five to six classes (*Chen et al., 2017*). This was followed by 2D classification to further remove ice, and another round of ab initio reconstruction searching for three classes with 620,611 particles. The best map reconstructed from 156,694 particles was refined to ~4.5 Å using non-uniform refinement, and then further improved to ~3.6 Å by Bayesian polishing and 3D refinement in Relion 3.1 (*Zivanov et al., 2018*; *Punjani et al., 2020*).

The final map generated from iterative ab initio reconstruction was then used with 7 decoy maps for sorting 3,908,602 particles by iterative heterogeneous refinement (*Chen et al., 2017*). The best particles from this procedure were refined, particle duplicates were removed, and the map was improved by Bayesian polishing, 3D refinement and 3D classification, searching for 3 classes with fixed angular distribution and T=40 parameters in Relion 3.1 (*Ritchie et al., 2009*). A subset of 15,848 particles was identified that generated a 3.6 Å reconstruction after 3D refinement. This stack of particles and map was then used to train the Topaz ResNet8 neural network to re-select particles, and for iterative heterogeneous refinement. This entire procedure was repeated twice more and the final map, which was selected from a 3D classification step in Relion 3.1, followed by CTF refinement, 3D refinement, Bayesian polishing and final 3D refinement to generate a reconstruction to 3.1 Å with 162,607 particles.

Data processing information for the SARS-CoV-2 Orf3a LE/Lyso MSP1D1 nanodisc, SARS-CoV-2 Orf3a PM MSP1D1 nanodisc, SARS-CoV-2 Orf3a LE/Lyso Saposin A nanodisc and SARS-CoV-1 Orf3a LE/Lyso MSP1D1 nanodisc datasets is summarized in *Supplementary file 1*.

## Model building and refinement

The atomic models were manually built into one of the half-maps that had been 'postprocessed' in Relion 3.0 or Relion 3.1 using a *B*-factor of −50 Å$^2$ and low-pass filtered at the final overall resolution, and the same half map that had been density-modified using resolve-cryo-EM in PHENIX (*Zivanov et al., 2018*; *Terwilliger et al., 2020*). The high-resolution structure of SARS CoV-2 Orf3a was used as a starting point (PDB ID: 7KJR) and refined in real space using the COOT software (*Emsley et al., 2010*). The atomic models were further refined in real space against the same half-map using real-space-refine-1.19 in PHENIX (*Liebschner et al., 2019*). The final models have good stereochemistry and good Fourier shell correlation with the other half-map as well as the combined map (*Figure 3—figure supplement 3*, *Figure 4—figure supplement 3*). Model refinement and validation statistics can be found in *Supplementary file 1*. Structural figures were prepared with Pymol (https://pymol.org/2/), Chimera, HOLE and APBS (*Pettersen et al., 2004*; *Smart et al., 1996*; *Jurrus et al., 2018*).

## Orf3a reconstitution into liposomes for functional assays

SEC-purified protein [in SEC buffer containing 3 mM DM (Anatrace)] was reconstituted into liposomes. For flux assays, a 3:1 (wt/wt) mixture of 1-palmitoyl-2-oleoyl-sn-glycero-3-phosphocholine (POPE, Avanti Polar Lipids) and 1-palmitoyl-2-oleoyl-sn-glycero-3-phospho-(1'-rac-glycerol POPG, Avanti Polar Lipids) was prepared at 20 mg/mL in reconstitution buffer (see flux assay protocols below). 8% (wt/vol) n-octyl-b-D-maltopyranoside (OM, Anatrace) was added to solubilize the lipids, and the mixture was incubated with rotation for 30 min at room temperature. Purified protein was mixed with an equal volume of the solubilized lipids to give a final protein concentration of ~0.08–0.8 mg/mL and a lipid concentration of 10 mg/mL (at ~1:100 or 1:10, wt:vol ratio). Proteoliposomes were formed by dialysis (using an 8000 Da molecular mass cutoff, Spectrum Labs) for 4–5 days at 4 °C against 4 L of reconstitution buffer and flash-cooled and stored at –80 °C until use. For all experiments, a liposome-only control was prepared concurrently in the absence of protein.

For proteoliposome patch-clamp, soybean polar lipid extract (Asoletin, Avanti Polar Lipids) was prepared at 10 mg/mL in reconstitution buffer including 200 mM KCl, 10 mM HEPES, pH 7.4. Purified protein was concentrated to ~0.08–0.8 mg/mL and mixed dropwise with an equal volume of asolectin (at ~1:100 or 1:10, wt:vol ratio), diluted ~5–10 fold in reconstitution buffer, and transferred to a dialysis cassette (10,000 Da molecular mass cutoff Slide-A-Lyzer, Thermo-Fisher). Proteoliposomes were formed by detergent dialysis for 4–5 days at 4 °C against 4 L of reconstitution buffer. After dialysis, proteoliposomes were concentrated to a final lipid concentration of 50 mg/mL by ultracentrifugation at 100,000 x $g$ for 1 hr at 4 °C, followed by resuspension and sonication in the appropriate volume of fresh reconstitution buffer. Samples were aliquoted, flash-cooled and stored at –80 °C. For all experiments, a liposome-only control was prepared concurrently in the absence of protein.

## ACMA fluorescence-based flux assay

The ACMA fluorescence-based flux assays were performed as described previously with modification (*Zhang et al., 1994*; *Heginbotham et al., 1998*; *Miller and Long, 2012*; *Kane Dickson et al., 2014*). Briefly, Orf3a was reconstituted into liposomes by dialysis in buffer containing 150 mM KCl and 10 mM HEPES, pH 7.0 (pH-adjusted with N-methyl-D-glucamine, NMDG) to evaluate $K^+$ flux, or 100 mM $Na_2SO_4$ and 10 mM HEPES, pH 7.0 (pH-adjusted with NaOH) to test for $Cl^-$ flux. Reconstituted Orf3a was quickly thawed, sonicated, and diluted 1:100 into Flux Assay Buffer (FAB) containing 150 mM NMDG-Cl, 10 mM HEPES, pH 7.0, 0.5 mg/mL BSA, and 2 µM 9-Amino-6-Chloro-2-Methoxyacridine (ACMA, Sigma-Aldrich) to establish a $K^+$ gradient. A similar FAB was prepared to create a $Cl^-$ gradient by substituting 150 mM NMDG-Cl with 125 mM NaCl. The assay was performed in a 96-well format using a CLARIOstar Plus microplate reader (BMG Labtech) with excitation and emission wavelengths set to 410 nm and 490 nm, respectively. Fluorescence was recoded in kinetics mode for 20 min with 30 s between each read. After 150 s, the proton ionophore carbonyl cyanide m-chlorophenyl hydrazone (CCCP, Sigma-Aldrich) was added at 1 µM to each sample to promote the uptake of protons ($H^+$) due to the electrochemical gradient established by efflux ($K^+$) or influx ($Cl^-$) of ions through Orf3a. At the end of the $K^+$ flux assay (930 s), 20 nM valinomycin (Sigma-Aldrich) was added to each sample to release all $K^+$ from vesicles.

## 90° light-scattering flux assay

The 90° light-scattering flux assay was adapted from a published protocol (*Brammer et al., 2014*; *Stockbridge et al., 2012*). Orf3a was reconstituted in dialysis buffer containing 200 mM K-glutamate and 20 mM HEPES, pH 7.2. Liposomes were thawed, sonicated, and diluted 1:200 into a hypertonic FAB containing 260 mM K-thiocyanate (KSCN) and 20 mM HEPES, pH 7.2, in a stirred cuvette. 90° scattered light was recorded for 10 min using a Varian Carey UV-Vis Spectrophotometer (Agilent) with excitation and emission wavelengths at 600 nm. After 480 s, 20 nM valinomycin (Sigma-Aldrich) was added to each sample to measure $K^+$ influx for all vesicles in the sample, resulting in vesicle swelling and a reduction in light scattering.

## Proteoliposome patch-clamp experiments

Excised proteoliposome voltage clamp recordings were performed similar to protocols previously described (*Kern et al., 2021*; *Hulse et al., 2018*). Briefly, purified SARS-CoV-2 Orf3a was reconstituted in soybean polar lipid extract (Avanti Polar Lipids) at 1:10 and 1:100 wt:wt protein to lipid ratios.

Samples of this reconstitution were then dried on a cover glass under continuous vacuum at 4 °C for 12 h. The dried aliquots were then rehydrated for 40–50 min with rehydration buffer: 200 mM KCl, and 5 mM HEPES at pH 7.2. A suspension of the rehydrated proteoliposomes was collected and used to seed a recording chamber cover glass. Once the proteoliposomes settled (15–20 minutes), multi- GΩ seals were achieved with SylGard coated pipettes pulled and polished to 2–4 MΩ. The excised patch configuration was then achieved by retracting the pipette and breaking the connecting membrane with a physical jolt to the head stage. For symmetric recording conditions, internal and external solutions were as follows (in mM): Sodium solution; 150 NaCl, 1 $MgCl_2$, 1 $CaCl_2$, and 5 HEPES: Potassium solution; 150 KCl, 1 $MgCl_2$, 1 $CaCl_2$, and 5 HEPES: Calcium solution; 100 $CaCl_2$, 1 $MgCl_2$, and 5 HEPES. Voltage-clamp recordings were performed using an Axopatch 200-B in patch configuration and the data acquired via Digidata 1440 A and Clampex 10.7 software (Molecular Devices). High resolution current-voltage step protocols were performed at 125 kHz sampling rate with a 5 kHz Bessel filter. Extended/time-course recordings were acquired using 1 s repeated step protocols with a 10 kHz sampling rate and 2 kHz Bessel filtering. Gap-free recordings were performed at 100 kHz with a 5 kHz Bessel filter. Analysis of single channel sweeps was performed using ClampSeg GUI idealization, as well as custom manual thresholding and delta-search scripts, to calculate the conductance and dwell time of all single channel events present (*Hotz et al., 2013*; *Pein et al., 2021*). Event probabilities were then calculated as the mean open probability of a given conductance across all successful excised patch recordings.

## Mass spectrometry

Mass spectrometry and data analysis of SARS-CoV-2 Orf3a vesicles reconstituted at a high protein to lipid ratio were performed by the Proteomics Core Facility at the Whitehead Institute (Boston, MA).

## Co-immunoprecipitation experiments

Each aliquot of cells expressing VPS39$_{GFP}$ were resuspended in 5 mL of buffer containing (in mM) 50 HEPES, pH 7.5, 75 NaCl, 75 KCl, 1 EDTA, pH 8.0, a 1:1000 dilution of PI Mix III and 0.5% NP-40 (Millipore Sigma). Samples were incubated on dry ice for 10–20 min to lyse and then divided into 800 µL aliquots. For binding with SARS-CoV-1 or SARS-CoV-2 Orf3a constructs, protein purified in DM (0.5–50 µg), or a DM-buffer control, was added to each sample and rotated for 1 hr at 4 °C. Each sample was centrifuged at 14,000 x *g* for 15 min to remove cell debris. Input samples for the co-IP were examined on a gel at this stage. The rest of the sample was incubated with 50 µL bed volume of StrepTactinXT for 2 hr at 4 °C and flow-through after binding collected. Resin was washed x5 with wash buffer containing (in mM): 50 HEPES, pH 7.5, 75 NaCl, 75 KCl, 1 EDTA, pH 8.0, and 0.05% NP-40. The third wash step was used as a wash control. Wash buffer was added, mixed 1:1 with 2x loading buffer (50% Laemmli Sample Buffer, 500 mM DTT) to the resin and collected.

The western blot protocol is similar to that described for the surface biotinylation experiments. Samples were separated on 4–20% Mini-PROTEAN TGX Precast Protein Gels 150 V for 60 min. For streptavidin detection, membranes were probed with mouse StrepMAB-Classic antibody diluted 1:5000 in TBS-T blocking buffer (TBS-T with 3% BSA), followed by horseradish peroxidase conjugated to anti-mouse secondary antibody in TBS-T blocking buffer and developed using SuperSignal West Pico PLUS Chemiluminescent Substrate. For GFP detection, membranes were probed with rabbit monoclonal GFP antibody diluted 1:4000 in TBS-T blocking buffer, followed by horseradish peroxidase conjugated anti-rabbit secondary antibody at 1:10,000 dilution in TBS-T blocking buffer and developed using SuperSignal West Pico PLUS Chemiluminescent Substrate.

## Cell lines

All cell lines were purchased from ATCC, and they verified the authenticity of the cell lines by STR. We routinely perform mycoplasma testing and these cells lines have been tested.

## Acknowledgements

We'd like to thank Chris Miller for consultation on ion channel reconstitutions and the 90° light-scattering flux assay and Luke Lavis for use of their Varian Carey UV-Vis Spectrophotometer for these experiments. We also thank Lukas Frey for helpful suggestions with MSP1D1 nanodisc preparations, Kathy Schaefer for providing her expertise in FACS sorting, Eric Spooner for assistance with

mass spectrometry, and William Patton of Scientific Computing at the Janelia Research Campus for assistance with single-channel recording analysis. We would also like to acknowledge cryo-EM facility staff members Momoko Shiozaki and Shixin Yang for providing technical and computational support. Finally, we would like to thank members of the Clapham Lab for productive discussion, and in particular Ray Hulse for insight into structural interpretation and comments on the manuscript, and Shu-Hsien Sheu for helpful suggestions with collection and processing of confocal imaging data. DC-B received funding from the Comisión Nacional de Investigación Científica y Tecnológica (21170800). BCA received funding from the Australian National Health and Medical Research Council CJ Martin Fellowship (APP1162427). The work in the Ren lab was supported, in part, by NIH grants 1 R01 GM133172 and 1 R01 HL147379 (to DR). SEB received funding from the Millennium Nucleus of Ion Channel- Associated Diseases (MiNICAD, #NCN19_168), a Millennium Nucleus of the Iniciativa Milenio, National Agency of Research and Development (ANID, Chile).

## Additional information

### Funding

| Funder | Grant reference number | Author |
|---|---|---|
| National Institutes of Health | GM133172 | Dejian Ren |
| National Health and Medical Research Council | APP1162427 | Ben Cristofori-Armstrong |
| Comisión Nacional de Investigación Científica y Tecnológica | 21170800 | Deny Cabezas-Bratesco |
| Millennium Nucleus of Ion Channels -- Associated Diseases | NCN19_168 | Sebastian E Brauchi |
| National Institutes of Health | HL147379 | Dejian Ren |

The funders had no role in study design, data collection and interpretation, or the decision to submit the work for publication.

### Author contributions

Alexandria N Miller, Conceptualization, Data curation, Formal analysis, Validation, Investigation, Visualization, Methodology, Writing – original draft, Writing – review and editing; Patrick R Houlihan, Data curation, Software, Formal analysis, Validation, Investigation, Visualization, Methodology, Writing – review and editing; Ella Matamala, Data curation, Formal analysis, Investigation, Writing – review and editing; Deny Cabezas-Bratesco, Ben Cristofori-Armstrong, Data curation, Formal analysis, Validation, Investigation, Visualization, Writing – review and editing; Gi Young Lee, Tanya L Dilan, Data curation, Formal analysis, Investigation, Visualization, Writing – review and editing; Silvia Sanchez-Martinez, Doreen Matthies, Rui Yan, Data curation, Investigation, Writing – review and editing; Zhiheng Yu, Dejian Ren, Supervision, Methodology, Writing – review and editing; Sebastian E Brauchi, Conceptualization, Data curation, Formal analysis, Supervision, Investigation, Visualization, Writing – review and editing; David E Clapham, Supervision, Funding acquisition, Project administration, Writing – review and editing

### Author ORCIDs

Alexandria N Miller http://orcid.org/0000-0003-3986-7923
Patrick R Houlihan http://orcid.org/0000-0002-2505-2347
Tanya L Dilan http://orcid.org/0000-0002-3944-8385
Doreen Matthies http://orcid.org/0000-0001-9221-4484
Sebastian E Brauchi http://orcid.org/0000-0002-8494-9912
David E Clapham http://orcid.org/0000-0002-4459-9428

**Decision letter and Author response**
Decision letter https://doi.org/10.7554/eLife.84477.sa1
Author response https://doi.org/10.7554/eLife.84477.sa2

## Additional files

### Supplementary files
• MDAR checklist
• Supplementary file 1. Cryo-EM data collection, refinement, and validation statistics.

### Data availability

All constructs and stable cell lines generated are available upon request. Atomic coordinates and cryo-EM density maps of have been deposited with the Protein Data Bank and Electron Microscopy Data Bank with the accession numbers: 8EQJ (SARS-CoV-2 Orf3a, LE/Lyso, MSP1D1 nanodisc; EMD-28538), 8EQT (SARS-CoV-2 Orf3a, PM, MSP1D1 nanodisc; EMD-28545), 8EQU (SARS-CoV-2 Orf3a, LE/Lyso, Saposin A nanodisc; EMD-28546) and 8EQS (SARS-CoV-1 Orf3a, LE/Lyso, MSP1D1 nanodisc; EMD-28544).

The following datasets were generated:

| Author(s) | Year | Dataset title | Dataset URL | Database and Identifier |
|---|---|---|---|---|
| Miller AN, Houlihan PR, Matamala E, Cabezas-Bratesco D, Lee GY, Cristofori-Armstrong B, Dilan TL, Sanchez-Martinez S, Matthies D, Yan R, Yu Z, Ren D, Brauchi SE, Clapham DE | 2023 | Structure of SARS-CoV-2 Orf3a in late endosome/lysosome-like membrane environment, MSP1D1 nanodisc | https://www.rcsb.org/structure/8EQJ | RCSB Protein Data Bank, 8EQJ |
| Miller AN, Houlihan PR, Matamala E, Cabezas-Bratesco D, Lee GY, Cristofori-Armstrong B, Dilan TL, Sanchez-Martinez S, Matthies D, Yan R, Yu Z, Ren D, Brauchi SE, Clapham DE | 2023 | Structure of SARS-CoV-2 Orf3a in late endosome/lysosome-like membrane environment, MSP1D1 nanodisc | https://www.ebi.ac.uk/emdb/EMD-28538 | Electron Microscopy Data Bank, EMD-28538 |
| Miller AN, Houlihan PR, Matamala E, Cabezas-Bratesco D, Lee GY, Cristofori-Armstrong B, Dilan TL, Sanchez-Martinez S, Matthies D, Yan R, Yu Z, Ren D, Brauchi SE, Clapham DE | 2023 | Structure of SARS-CoV-2 Orf3a in plasma membrane-like environment, MSP1D1 nanodisc | https://www.rcsb.org/structure/8EQT | RCSB Protein Data Bank, 8EQT |
| Miller AN, Houlihan PR, Matamala E, Cabezas-Bratesco D, Lee GY, Cristofori-Armstrong B, Dilan TL, Sanchez-Martinez S, Matthies D, Yan R, Yu Z, Ren D, Brauchi SE, Clapham DE | 2023 | Structure of SARS-CoV-2 Orf3a in plasma membrane-like environment, MSP1D1 nanodisc | https://www.ebi.ac.uk/emdb/EMD-28545 | Electron Microscopy Data Bank, EMD-28545 |

*Continued on next page*

*Continued*

| Author(s) | Year | Dataset title | Dataset URL | Database and Identifier |
|---|---|---|---|---|
| Miller AN, Houlihan PR, Matamala E, Cabezas-Bratesco D, Lee GY, Cristofori-Armstrong B, Dilan TL, Sanchez-Martinez S, Matthies D, Yan R, Yu Z, Ren D, Brauchi SE, Clapham DE | 2023 | Structure of SARS-CoV-2 Orf3a in late endosome/lysosome-like environment, Saposin A nanodisc | https://www.rcsb.org/structure/8EQU | RCSB Protein Data Bank, 8EQU |
| Miller AN, Houlihan PR, Matamala E, Cabezas-Bratesco D, Lee GY, Cristofori-Armstrong B, Dilan TL, Sanchez-Martinez S, Matthies D, Yan R, Yu Z, Ren D, Brauchi SE, Clapham DE | 2023 | Structure of SARS-CoV-2 Orf3a in late endosome/lysosome-like environment, Saposin A nanodisc | https://www.ebi.ac.uk/emdb/EMD-28546 | Electron Microscopy Data Bank, EMD-28546 |
| Miller AN, Houlihan PR, Matamala E, Cabezas-Bratesco D, Lee GY, Cristofori-Armstrong B, Dilan TL, Sanchez-Martinez S, Matthies D, Yan R, Yu Z, Ren D, Brauchi SE, Clapham DE | 2023 | Structure of SARS-CoV-1 Orf3a in late endosome/lysosome-like environment, MSP1D1 nanodisc | https://www.rcsb.org/structure/8EQS | RCSB Protein Data Bank, 8EQS |
| Miller AN, Houlihan PR, Matamala E, Cabezas-Bratesco D, Lee GY, Cristofori-Armstrong B, Dilan TL, Sanchez-Martinez S, Matthies D, Yan R, Yu Z, Ren D, Brauchi SE, Clapham DE | 2023 | Structure of SARS-CoV-1 Orf3a in late endosome/lysosome-like environment, MSP1D1 nanodisc | https://www.ebi.ac.uk/emdb/EMD-28544 | Electron Microscopy Data Bank, EMD-28544 |

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
