## [Editor Report]

The function of specific proteins made by SARS-CoV-1 and SARS-CoV-2 is under debate, with diverging claims previously published regarding the ability of Orf3a proteins from either virus to form ion channels. The authors undertook a thorough characterization of Orf3a from CoV-1 and CoV-2 by combining data from a range of different structural and functional experiments, arguably providing the most compelling evidence to date that Orf3a from viruses is not an ion channel. Instead, the orthologue-specific interaction with a component of a larger protein complex suggests the role of one of the two membrane proteins in the endo-lysosomal pathway. The work is significant from a fundamental science perspective, for its implications for COVID antiviral development strategies, and also for establishing guidelines for future identification of true viral ion channels.

---

## [Decision Letter]

**Decision letter after peer review:**

Thank you for submitting your article "The SARS-CoV-2 accessory protein Orf3a is not an ion channel, but does interact with trafficking proteins" for consideration by *eLife*. Your article has been reviewed by 3 peer reviewers, and the evaluation has been overseen by a Reviewing Editor and Kenton Swartz as the Senior Editor. The following individuals involved in review of your submission have agreed to reveal their identity: Youxing Jiang (Reviewer #1); Raimund Dutzler (Reviewer #3).

Essential revisions:

The three reviewers were enthusiastic and put forward a number of suggestions for improving the manuscript in revision. The following are the most important points to focus your efforts.

1) It would be valuable to provide a more detailed comparison of the determined structures with previous structures of SARS-CoV-2 ORF3a and to consider whether there are similarities to other proteins of known structure and function that might provide further insight into potential functional properties of ORF3a.

2) The quality of the presumed lipid density and its interaction with the protein is difficult to judge and its presentation should be improved. Although of potential relevance, the discussion of the lipid binding site seems exaggerated. The presence of lipids in a membrane protein is not unexpected and not every bound lipid is of functional importance. If you have additional evidence for the importance of the bound lipid, we encourage you to include that in the manuscript.

3) The interaction of MSP1D1 and Saposin A with the protein is somewhat unusual and we wondered whether there is anything that can be learned with respect to ORF3a function.

4) Please provide the information requested by reviewer #2 concerning reproducibility for some experiments and in one case a missing control.

*Reviewer #2 (Recommendations for the authors):*

1. Please include a control for Figure 1C showing a non-induced (non-Orf3a-expressing) cell.

2. Figure 1 – how many images/independent experiments are the images representative of? Please state in the legend.

3. Figure 2B – are these mean or exemplar traces? Please specify in the legend.

4. Figure 6 – how many independent experiments are the western blots/co-IPs representative of? Please specify in the legend.

*Reviewer #3 (Recommendations for the authors):*

The general quality of performed experiments is very high and I have only few remarks the authors should address.

• I think that a detailed comparison of the determined structures with previous structures of SARS-CoV-2 ORF3a should already be provided in the results. It would also be interesting whether the similarity to other proteins with a similar structure could provide any insight into potential functional properties.

• The quality of the presumed lipid density and its interaction with the protein is difficult to judge and its presentation should be improved.

• Although of potential relevance, I find the discussion of the lipid binding site exaggerated. The presence of lipids in a membrane protein is not unexpected and not every bound lipid is of functional importance. In case there is additional evidence, it should be provided.

• The interaction of MSP1D1 and Saposin A with the protein is unusual. I wonder whether there is anything that can be learned with respect to ORF3a function.

---

## [Author Response]

Essential revisions:The three reviewers were enthusiastic and put forward a number of suggestions for improving the manuscript in revision. The following are the most important points to focus your efforts.1) It would be valuable to provide a more detailed comparison of the determined structures with previous structures of SARS-CoV-2 ORF3a and to consider whether there are similarities to other proteins of known structure and function that might provide further insight into potential functional properties of ORF3a.

Our cryo-EM structures of SARS-CoV-2 Orf3a are nearly identical with the previously determined structures of SARS-CoV-2 Orf3a. In the discussion, we have clarified this by adding the global RMSD (0.38 Å) between our SARS-CoV-2 Orf3a PM MSP1D1 nanodisc and the previously published SARS-CoV-2 Orf3a structures in the text, and we also included a figure of our PM nanodisc model aligned with the 2.1 Å resolution published model (see *Figure 3—figure supplement 3K*).

In Kern et al., 2021, the authors also describe a tetrameric, low resolution (6.5 Å) cryo-EM structure of SARS-CoV-2 Orf3a.^1^ We also observe a SARS-CoV-2 Orf3a protein population with a larger hydrodynamic radius by size-exclusion chromatography, similar to Kern et al. (pink trace, “4 mer”, Author response image 1).^1^ However, we suspect that this oligomer may not represent a native assembly of the protein. We have yet to observe a SARS-CoV-2 Orf3a tetramer in isolated cell membranes by chemical crosslinking with either bismaleimidohexane (*Figure 3—figure supplement 3I*) or disuccinimidyl suberate (Miller and Clapham, unpublished). The cryo-EM structure of the SARS-CoV-2 Orf3a tetramer is formed by two Orf3a dimers. The point of interaction between the dimers is minimal, mediated by a cytosolic loop between β1-2. Among the residues within this loop is C153, which we suspect might be forming a non-specific disulfide bond during purification. In support of this, C153 is not present in SARS-CoV-1 Orf3a and, accordingly, we do not see a SARS-CoV-1 Orf3a protein population with a larger hydrodynamic radius by size-exclusion chromatography when purified in the same buffer and detergent conditions as SARS-CoV-2 Orf3a (green trace, “no 4 mer”, Author response image 1). Although more work would need to be done to convince ourselves on this matter, these lines of evidence begin to suggest that the SARS-CoV-2 Orf3a tetramer is formed by a non-specific disulfide bond during purification, and that it may not be physiologically relevant.

**Author response image 1. sa2fig1:** 

A distant relative of Orf3a proteins of coronaviruses is the M (membrane) protein, which is the most abundant protein in the virion and is critical for viral assembly and membrane budding.^,2,3^ The M protein of SARS-CoV-1 was reported to assemble as a dimer and to adopt several conformations, a long and compact form, in the SARS-CoV-1 virion and in viral-like particles as visualized by low-resolution cryo-EM (>10 Å resolution) and cryo-ET.^4^ The M protein has also been shown to interact with the RNA-binding N (nucleocapsid) protein in SARS-CoV-1 and SARS-CoV-2.^5-8^ Recently, three groups independently solved high-resolution X-ray and cryo-EM structures of dimeric M proteins from SARS-CoV-2 and a bat coronavirus, which have an overall fold that is shared with the Orf3a proteins.^9-11^ In two of these publications, the authors also observe the long and compact conformations of the M protein, supporting the previous findings for the SARS-CoV-1 M protein.^4, 10,11^ Neither of these structures has a clear ion pore, and the M protein has not been reported to exhibit viroporin activity.^10,11^ It is unclear what these conformations represent functionally. It was suggested from the SARS-CoV-1 M protein tomography studies that its long conformation may enforce rigidity and membrane curvature of the virion, which is ~100 nm diameter on average, and may facilitate binding to the N protein or direct interaction with viral RNA.^4^ The compact form is found in more elongated/flattened areas of the membrane, and its endodomain does not appear to contact the N protein and viral DNA.^4^

Superposition of SARS-CoV-2 Orf3a with the long and compact structures of the M protein yields an RMSD of 3.7 and 4.4 Å, respectively.^10^ Interestingly, the 50 Å diameter of the SARS-CoV-2 Orf3a homodimer is more in line with the diameter of M protein long conformation (50 Å) than the compact conformation (57 Å). We therefore wonder whether the totality of this evidence – the structural homology of Orf3a proteins with M protein and its similarity in diameter to the M protein long conformation, the proposed lipid binding sites that we observe in the Orf3a protein structures, and the reported function of SARS-CoV1 Orf3a to promote the formation of intracellular vesicles – could be consistent with a contribution to intracellular membrane remodeling.^12^ Furthermore, Golgi fragmentation is another cellular hallmark of SARS-CoV-1 Orf3a overexpression, a phenotype which is shared with SARS-CoV-2 Orf3a based on our preliminary data (Matamala and Brauchi, unpublished) and a recent publication.^12,13^ Overall, although the membrane remodeling hypothesis is intriguing, it is highly speculative and for this reason we have chosen to exclude this discussion from the manuscript.

2) The quality of the presumed lipid density and its interaction with the protein is difficult to judge and its presentation should be improved. Although of potential relevance, the discussion of the lipid binding site seems exaggerated. The presence of lipids in a membrane protein is not unexpected and not every bound lipid is of functional importance. If you have additional evidence for the importance of the bound lipid, we encourage you to include that in the manuscript.

Good point! We reinspected our Figures and we agree that the presentation of the lipid density needed to be improved. We have therefore modified Figure 4 to include DOPE molecules in Lipid Sites 1 (orange) and 2 (purple) in Figure 4A for visual clarity, and to be consistent with Figures 4B and C. In addition, we enhanced the contour level of the putative lipid electron density in Figure 4A-C to 7s from 2.5s and this significantly reduced the noise and improved visualization. In Figure 4D-I, we thickened the lines representing bonds for the DOPE molecules and side chains for clarity. In addition, after generating this figure for initial submission, we modified the Orf3a structures for PDB deposition by trimming the lipid acyl chains for more accurate representation. These updated models should have been used to generate the images in this Figure, and in this revision, we have therefore regenerated all the panels in Figure 4 with the PDB deposited models. These minor modifications do not change the interpretation of the data.

The functional significance of these lipids is unclear, and as the reviewer pointed out, the presence of lipids associated with membrane proteins is common. Although we believe that the lipids might have a physiological role and this could hint at a general mechanism of Orf3a protein function (see discussion of the M protein above), we agree that this may have been overinterpreted and we have therefore qualified some of the more speculative conclusions.

3) The interaction of MSP1D1 and Saposin A with the protein is somewhat unusual and we wondered whether there is anything that can be learned with respect to ORF3a function.

The membrane scaffold interaction appears to be shared between SARS-CoV-1 and SARS-CoV-2 Orf3a. Could this interaction be physiologically relevant and help us narrow our search for a common function of Orf3a proteins? We have wondered if this region of Orf3a could be an interaction hub for membrane tethering and/or integral membrane proteins. A known host membrane protein that was originally identified by mass spectrometry to interact with both SARS-CoV-1 and SARS-CoV-2 Orf3a proteins is CLCC1 (CLIC-like protein).^14,15^ We have experimentally validated this by co-IP (Miller, Sanchez-Martinez, and Clapham, unpublished), and others have published a similar finding.^16^ CLCC1 is an ER-resident integral membrane protein that is annotated to be a chloride channel, CLIC-like 1, but as is the case with the channel function of Orf3a proteins, the experimental evidence for this is quite weak in our opinion.^17^ Furthermore, there is no atomic resolution structure of CLCC1. Although these data begin to hint at a general function for Orf3a proteins, we agree that the interaction hub hypothesis is highly speculative and the function of CLCC1 is unclear, so we chose to not include this discussion in the manuscript.

4) Please provide the information requested by reviewer #2 concerning reproducibility for some experiments and in one case a missing control.

We have addressed this by including a non-induced Orf3a_HALO_ control (Figure 1—figure supplement 1B) and have replaced Figure 1C with an image showing plasma membrane localization of Orf3a_HALO_ by TIRF (the original image was of Orf3a_SNAP_). The Figure legends and text have also been updated to reflect these changes. All other points that were mentioned by reviewer #2 have been addressed and added to the Figure legends.

Citations:

1) Kern DM, Sorum B, Mali SS, Hoel CM, Sridharan S, Remis JP, Toso DB, Kotecha A, Bautista DM, Brohawn SG. 2021. Cryo-EM structure of SARS-CoV-2 ORF3a in lipid nanodiscs. *Nature Structural & Molecular Biology*
**28**:573–582. doi:10.1038/s41594-021-00619-0, PMID 34158638

2) Siu YL, Teoh KT, Lo J, Chan CM, Kien F, Escriou N, Tsao W, Nicholls JM, Altmeyer, JS, Peiris, M, Bruzzone, R, Nal, B. 2008. The M, E and N structural proteins of the severe acute respiratory syndrome coronavirus are required for efficient assembly, trafficking, and release of virus-like particles. *Journal of Virology*
**82**:11318–11330. doi:10.1128/JVI.01052-08, PMID:18753196

3) Bar-On YM, Flamholz A, Phillips R, Milo R. 2020. SARS-CoV-2 (COVID-19) by the numbers. *eLife*
**9**:e57309. doi:10.7554/eLife.57309, PMID:32228860

4) Neuman BW, Kiss G, Kunding AH, Bhella D, Baksh MF, Connelly S, Droese B, Klaus JP, Makino S, Sawicki SG, Siddell SG, Stamou DG, Wilson IA, Kuhn P, Buchmeier MJ. 2011. A structural analysis of M protein in coronavirus assembly and morphology. *Journal of Structural Biology*
**174**:11–22. doi:10.1016/j.jsb.2010.11.021, PMID:21130884

5) Narayanan K, Maeda A, Maeda J, Makino S. 2000. Characterization of the coronavirus M protein and nucleocapsid interaction in infected cells. *Journal of Virology*
**74**:8127–8134. doi:10.1128/JVI.74.17.8127-8134.2000, PMID:10933723

6) Hurst KR, Kuo L, Koetzner CA, Ye R, Hsue B, Masters PS. 2005. A major determinant for membrane protein interaction localizes to the carboxy-terminal domain of the mouse coronavirus nucleocapsid protein. *Journal of Virology*
**79**:13285–13297. doi:10.1128/JVI.79.21.13285-13297.2005, PMID:16227251

7) Kuo L, Hurst-Hess KR, Koetzner CA, Masters PS. 2016. Analyses of coronavirus assembly interactions with interspecies membrane and nucleocapsid protein chimeras. *Journal of Virology*
**90**:4357–4368. doi:10.1128/JVI.03212-15, PMID:26889024

8) Lu S, Ye Q, Singh D, Cao Y, Diedrich JK, Yates JR, Villa E, Cleveland DW, Corbett KD. 2021. The SARS-CoV-2 nucleocapsid phosphoprotein forms mutually exclusive condensates with RNA and the membrane-associated M protein. *Nature Communications*
**12**:502. doi:10.1038/s41467-020-20768-y, PMID:33479198

9) Dolan KA, Dutta M, Kern DM, Kotecha A, Voth GA, Brohawn SG. 2022. Structure of SARS-CoV-2 M protein in lipid nanodiscs. *eLife*
**11**:e81702. doi:10.7554/eLife.81702, PMID:36264056

10) Zhang Z, Nomura N, Muramoto Y, Ekimoto T, Uemura T, Liu K, Yui M, Kono N, Aoki J, Ikeguchi M, Noda T, Iwata S, Ohto U, Shimizu T. 2022. Structure of SARS-CoV-2 membrane protein essential for virus assembly. *Nature Communications*
**13**:4399. doi:10.1038/s41467-022-32019-3, PMID:35931673

11) Wang X, Yang Y, Sun Z, Zhou X. 2022. Crystal structure of the membrane (M) protein from a SARS-CoV-2-related coronavirus,(*BioRxiv)*. doi:10.1101/2022.06.28.497981

12) Freundt EC, Yu L, Goldsmith CS, Welsh S, Cheng A, Yount B, Liu W, Frieman MB, Buchholz UJ, Screaton GR, Lippincott-Schwartz J, Zaki SR, Xu X-N, Baric RS, Subbarao K, Lenardo MJ. 2010. The open reading frame 3a protein of severe acute respiratory syndrome-associated coronavirus promotes membrane rearrangement and cell death. *Journal of Virology*
**84**:1097–1109. doi:10.1128/JVI.01662-09, PMID:19889773

13) Arshad N, Laurent-Rolle M, Ahmed WS, Hsu JC-C, Mitchell SM, Pawlak J, Sengupta D, Biswas KH, Cresswell P. 2023. SARS-CoV-2 accessory proteins ORF7a and ORF3a use distinct mechanisms to down-regulate MHC-I surface expression. *Proc Natl Acad Sci USA*
**120**:e2208525120. doi:10.1073/pnas.2208525120, PMID:35611331

14) Gordon DE, Hiatt J, Bouhaddou M, Rezelj VV, Ulferts S, Braberg H, Jureka AS, Obernier K, Guo JZ, Batra J, Kaake RM, Weckstein AR, Owens TW, Gupta M, Pourmal S, Titus EW, Cakir M, Soucheray M, McGregor M, Cakir Z, Jang G, O’Meara MJ, Tummino TA, Zhang Z, Foussard H, Rojc A, Zhou Y, Kuchenov D, Hüttenhain R, Xu J, Eckhardt M, Swaney DL, Fabius JM, Ummadi M, Tutuncuoglu B, Rathore U, Modak M, Haas P, Haas KM, Naing ZZC, Pulido EH, Shi Y, Barrio-Hernandez I, Memon D, Petsalaki E, Dunham A, Marrero MC, Burke D, Koh C, Vallet T, Silvas JA, Azumaya CM, Billesbølle C, Brilot AF, Campbell MG, Diallo A, Dickinson MS, Diwanji D, Herrera N, Hoppe N, Kratochvil HT, Liu Y, Merz GE, Moritz M, Nguyen HC, Nowotny C, Puchades C, Rizo AN, Schulze-Gahmen U, Smith AM, Sun M, Young ID, Zhao J, Asarnow D, Biel J, Bowen A, Braxton JR, Chen J, Chio CM, Chio US, Deshpande I, Doan L, Faust B, Flores S, Jin M, Kim K, Lam VL, Li F, Li J, Li Y-L, Li Y, Liu X, Lo M, Lopez KE, Melo AA, Moss FR, Nguyen P, Paulino J, Pawar KI, Peters JK, Pospiech TH, Safari M, Sangwan S, Schaefer K, Thomas PV, Thwin AC, Trenker R, Tse E, Tsui TKM, Wang F, Whitis N, Yu Z, Zhang K, Zhang Y, Zhou F, Saltzberg D, Hodder AJ, Shun-Shion AS, Williams DM, White KM, Rosales R, Kehrer T, Miorin L, Moreno E, Patel AH, Rihn S, Khalid MM, Vallejo-Gracia A, Fozouni P, Simoneau CR, Roth TL, Wu D, Karim MA, Ghoussaini M, Dunham I, Berardi F, Weigang S, Chazal M, Park J, Logue J, McGrath M, Weston S, Haupt R, Hastie CJ, Elliott M, Brown F, Burness KA, Reid E, Dorward M, Johnson C, Wilkinson SG, Geyer A, Giesel DM, Baillie C, Raggett S, Leech H, Toth R, Goodman N, Keough KC, Lind AL, Klesh RJ, Hemphill KR, Carlson-Stevermer J, Oki J, Holden K, Maures T, Pollard KS, Sali A, Agard DA, Cheng Y, Fraser JS, Frost A, Jura N, Kortemme T, Manglik A, Southworth DR, Stroud RM, Alessi DR, Davies P, Frieman MB, Ideker T, Abate C, Jouvenet N, Kochs G, Shoichet B, Ott M, Palmarini M, Shokat KM, García-Sastre A, Rassen JA, Grosse R, Rosenberg OS, Verba KA, Basler CF, Vignuzzi M, Peden AA, Beltrao P, Krogan NJ, QCRG Structural Biology Consortium, Zoonomia Consortium. 2020a. Comparative host-coronavirus protein interaction networks reveal pan-viral disease mechanisms. *Science*
**370**:eabe9403. doi:10.1126/science.abe9403, PMID:33060197

15) Gordon DE, Jang GM, Bouhaddou M, Xu J, Obernier K, White KM, O’Meara MJ, Rezelj VV, Guo JZ, Swaney DL, Tummino TA, Hüttenhain R, Kaake RM, Richards AL, Tutuncuoglu B, Foussard H, Batra J, Haas K, Modak M, Kim M, Haas P, Polacco BJ, Braberg H, Fabius JM, Eckhardt M, Soucheray M, Bennett MJ, Cakir M, McGregor MJ, Li Q, Meyer B, Roesch F, Vallet T, Mac Kain A, Miorin L, Moreno E, Naing ZZC, Zhou Y, Peng S, Shi Y, Zhang Z, Shen W, Kirby IT, Melnyk JE, Chorba JS, Lou K, Dai SA, Barrio-Hernandez I, Memon D, Hernandez-Armenta C, Lyu J, Mathy CJP, Perica T, Pilla KB, Ganesan SJ, Saltzberg DJ, Rakesh R, Liu X, Rosenthal SB, Calviello L, Venkataramanan S, Liboy-Lugo J, Lin Y, Huang XP, Liu Y, Wankowicz SA, Bohn M, Safari M, Ugur FS, Koh C, Savar NS, Tran QD, Shengjuler D, Fletcher SJ, O’Neal MC, Cai Y, Chang JCJ, Broadhurst DJ, Klippsten S, Sharp PP, Wenzell NA, Kuzuoglu-Ozturk D, Wang HY, Trenker R, Young JM, Cavero DA, Hiatt J, Roth TL, Rathore U, Subramanian A, Noack J, Hubert M, Stroud RM, Frankel AD, Rosenberg OS, Verba KA, Agard DA, Ott M, Emerman M, Jura N, von Zastrow M, Verdin E, Ashworth A, Schwartz O, d’Enfert C, Mukherjee S, Jacobson M, Malik HS, Fujimori DG, Ideker T, Craik CS, Floor SN, Fraser JS, Gross JD, Sali A, Roth BL, Ruggero D, Taunton J, Kortemme T, Beltrao P, Vignuzzi M, García-Sastre A, Shokat KM, Shoichet BK, Krogan NJ. 2020b. A SARS-CoV-2 protein interaction map reveals targets for drug repurposing. *Nature*
**583**:459–468. doi:10.1038/s41586-020-2286-9, PMID:32353859

16) Chen Z, Wang C, Feng X, Nie L, Tang M, Zhang H, Xiong Y, Swisher SK, Srivastava M, Chen J. 2021. Interactomes of SARS-CoV-2 and human coronaviruses reveal host factors potentially affecting pathogenesis. *The EMBO Journal*
**40**. doi:10.15252/embj.2021107776, PMID:34232536

17) Nagasawa M, Kanzaki M, Iino Y, Morishita Y, Kojima I. 2001. Identification of a novel chloride channel expressed in the endoplasmic reticulum, Golgi apparatus, and nucleus. *Journal of Biological Chemistry*
**276**:20413–20418. doi:10.1074/jbc.M100366200, PMID:11270957